# Selective targeting of ligand-dependent and -independent signaling by GPCR conformation-specific anti-US28 intrabodies

Timo W. M. De Groof [1,2,9], Nick D. Bergkamp[1,9], Raimond Heukers [1,3], Truc Giap[1], Maarten P. Bebelman[1], Richard Goeij-de Haas[4], Sander R. Piersma[4], Connie R. Jimenez[4], K. Christopher Garcia [5,6,7], Hidde L. Ploegh [8], Marco Siderius [1] & Martine J. Smit [1✉]

While various GPCRs, including US28, display constitutive, ligand-independent activity, it remains to be established whether ligand-dependent and -independent active conformations differ and can be selectively modulated. Previously, the agonist-bound conformation of US28 was stabilized and its structure was solved using the anti-US28 nanobody Nb7. Here we report the recognition of the constitutively active, apo-conformation of US28 by another nanobody VUN103. While the Nb7 intrabody selectively inhibits ligand-induced signaling, the VUN103 intrabody blocks constitutive signaling, indicating the existence of distinct US28 conformational states. By displacing Gα_q protein, VUN103 prevents US28 signaling and reduces tumor spheroids growth. Overall, nanobodies specific for distinct GPCR conformational states, i.e. apo- and agonist-bound, can selectively target and discern functional consequences of ligand-dependent versus independent signaling.

[1] Amsterdam Institute of Molecular and Life Sciences (AIMMS), Division of Medicinal Chemistry, Faculty of Sciences, VU University, Amsterdam, The Netherlands. [2] Department of Medical Imaging, In Vivo Cellular and Molecular Imaging Laboratory (ICMI), Vrije Universiteit Brussel (VUB), Brussels, Belgium. [3] QVQ B.V., Utrecht, The Netherlands. [4] Department of Medical Oncology, Amsterdam University Medical Center, VU University, Cancer Center Amsterdam, Amsterdam, The Netherlands. [5] Department of Molecular and Cellular Physiology, Stanford University School of Medicine, Stanford, USA. [6] Department of Structural Biology, Stanford University School of Medicine, Stanford, USA. [7] Howard Hughes Medical Institute, Stanford University School of Medicine, Stanford, USA. [8] Program in Cellular and Molecular Medicine, Boston Children's Hospital, Boston, USA. [9] These authors contributed equally: Timo W. M. De Groof, Nick D. Bergkamp. ✉email: mj.smit@vu.nl

G protein-coupled receptors (GPCRs) are a superfamily of receptors that regulate a variety of important physiological processes[1,2]. Although this superfamily consists of more than 800 genes divided into three main classes (class A, B, and C), their ability to engage in different modes of signaling, involving G-protein activation and β-arrestin recruitment, is highly conserved[2,3]. The binding of different ligands (agonist, biased agonist, antagonist, or inverse agonist) to these receptors induces specific conformational changes, which differentially modulate the downstream signaling network[3,4]. Some GPCRs, such as the human calcitonin receptor, human cannabinoid CB1 receptor, ghrelin receptor, melanocortin receptors, and in particular the viral GPCR US28, also signal in a ligand-independent manner[5–9]. How different ligands stabilize distinct GPCR conformations and modulate the ensuing signaling pathways is still not fully understood. In the past decade, nanobodies have proven to be invaluable tools for GPCR research. Nanobodies are antibody fragments derived from the variable region of heavy-chain only antibodies found in Camelids[10]. Nanobodies are small (~15 kDa), have a convex and relatively large paratope, and often a long protruding CDR3 region, which enables them to target epitopes less accessible to conventional antibodies[11,12]. Nanobodies that target either extracellular or intracellular epitopes on GPCRs have been reported[12,13]. When expressed intracellularly, they are referred to as intrabodies. In particular, nanobodies directed against the intracellular portion of GPCRs have provided insight in distinct GPCR conformations imposed by the presence of either antagonists or agonists. Most of these nanobodies recognize and stabilize specific GPCR conformations[12,13]. By stabilizing flexible receptor structures, nanobodies have served as crystallization chaperones, as conformational biosensors, and as tools in drug discovery[14–22].

A prototypical example of a constitutively active GPCR that can be further activated by ligand binding is US28. US28 is a chemokine receptor encoded by human cytomegalovirus (HCMV). It plays a role in cancer progression[23,24]. US28 binds various chemokines, including CCL5 and CX3CL1, and activates multiple signaling pathways, both in a ligand-independent and -dependent manner[25–30]. In its apo-state, US28 signals constitutively via $G\alpha_q$, while ligand binding can superactivate the receptor by signaling via $G\alpha_q$ and $G\alpha_{12/13}$[31,32]. Most receptor modulators bind to the extracellular portion of US28 and inhibit its activity by acting as antagonists and/or inverse agonists[24,32–36]. Based on these activity profiles, the ligand-bound active conformation of US28 might differ from its constitutively active apo-conformation. The crystal structure of US28 in complex with its ligand CX3CL1 was solved using a nanobody (Nb7) that stabilizes the intracellular portion of US28[37]. Nb7 as well as Nb11, which stabilizes a partially non-overlapping ligand-bound conformational state, have provided more insight into the promiscuity with which US28 recognizes chemokine ligands[38].

In this study, we develop and identify a nanobody that binds to the intracellular portion of US28 (VUN103). We map the binding site of VUN103 and evaluate the effect of VUN103 binding on US28 signaling. Using chemokine receptor binding, bioluminescence resonance energy transfer (BRET)-based assays, mass spectrometry, and confocal microscopy, we show that VUN103 and Nb7 differentially affect the conformation and signaling of US28. Nb7 stabilizes the ligand-bound, active conformation of US28 and inhibits ligand-induced signaling. VUN103 recognizes and inhibits a different, constitutively active, apo-conformation of US28. Our study thus demonstrates the existence of different conformational states of US28: the constitutively active conformation of this GPCR is distinct from the active conformation adopted upon chemokine binding. We demonstrate the functional impact of capturing US28's constitutively active form by

showing that VUN103 inhibits US28-mediated 3D growth of tumor spheroids. These nanobodies not only serve as tool compounds, but they could also distinguish between ligand-dependent and -independent US28 signaling and might thus guide the development of future therapeutics.

## Results

**VUN103 is a nanobody that binds to the intracellular portion of US28**. To identify nanobodies that target the constitutively active conformation of US28, a previously generated library of nanobodies from immunized llamas was used for panning on US28-expressing membranes[39]. Master plates were assessed for antigen binding in phage-ELISA. Hits were classified into families based on the amino acid sequence of their CDR3 region (Fig. 1a). One large family and multiple smaller families of US28-specific nanobodies were found. Multiple hits from the large family of nanobody clones were produced as recombinant proteins and further characterized and classified based on their binding affinity to US28. One lead hit (VUN103) of this family was selected based on its high affinity. Equilibrium binding ELISA with VUN103 indicated an apparent binding affinity of $22 \pm 4$ nM to US28 (Fig. 1b). To determine whether VUN103 bound an extracellular or intracellular epitope, we determined the ability of VUN103 to bind to intact and/or permeabilized US28-expressing cells. We visualized such binding by immunofluorescence microscopy. US28 expression was detected using a conventional anti-US28 antibody that targets the intracellular portion of US28 (Fig. 1c)[40]. VUN103 bound only to permeabilized but not to intact US28-expressing cells. VUN103 thus recognizes an intracellular epitope (Fig. 1d). In contrast, VUN100, which targets the extracellular portion of US28, bound to both intact and permeabilized US28-expressing cells, as expected. VUN103 did not bind to permeabilized US28-negative cells or cells expressing CX3CR1, the human chemokine receptor most homologous to US28 (Supplementary Fig. 1). To confirm that VUN103 binds to an intracellular of US28, we determined whether VUN103 was able to compete with US28 ligands (Fig. 1e, f). In contrast to the known extracellularly binding US28 nanobody VUN100 [39], VUN103 did not displace radiolabeled US28 ligands CCL5 and CX3CL1.

**VUN103 inhibits constitutive US28 signaling by G protein displacement**. Next, the functional consequences of VUN103 binding on the constitutive activity of US28 were determined. We assessed US28-mediated $G\alpha_q$ signaling by measuring inositol phosphate accumulation in US28-expressing cells, co-transfected with increasing amounts of DNA encoding VUN103-mVenus or a non-US28 targeting/irrelevant Nb-mVenus (referred now to as intrabodies). Increasing the expression levels of the VUN103 intrabody completely inhibited US28-mediated inositol phosphate accumulation, not seen for the non-US28 targeting intrabody (Fig. 2a). We obtained the same result with untagged intrabodies, indicating that mVenus-tagging of the intrabodies did not affect VUN103-mediated inhibition. In contrast, no functional consequences of VUN103-mVenus expression were observed on histamine-induced histamine H1 receptor-mediated inositol phosphate accumulation. We conclude that the functional impact of the VUN103 intrabody is US28-specific (Supplementary Fig. 2). Similar to the inositol phosphate accumulation assays, the VUN103 intrabody inhibited US28-mediated $G\alpha_q$ signaling via NFAT, STAT3, and NF-κB[40,41] (Fig. 2b–d).

Next, we compared the functional consequences of VUN103 intrabody expression on US28 constitutive activity with that of the Nb7 intrabody (Fig. 2e). Different intrabody-mVenus constructs were expressed at similar levels (Supplementary Fig. 3).

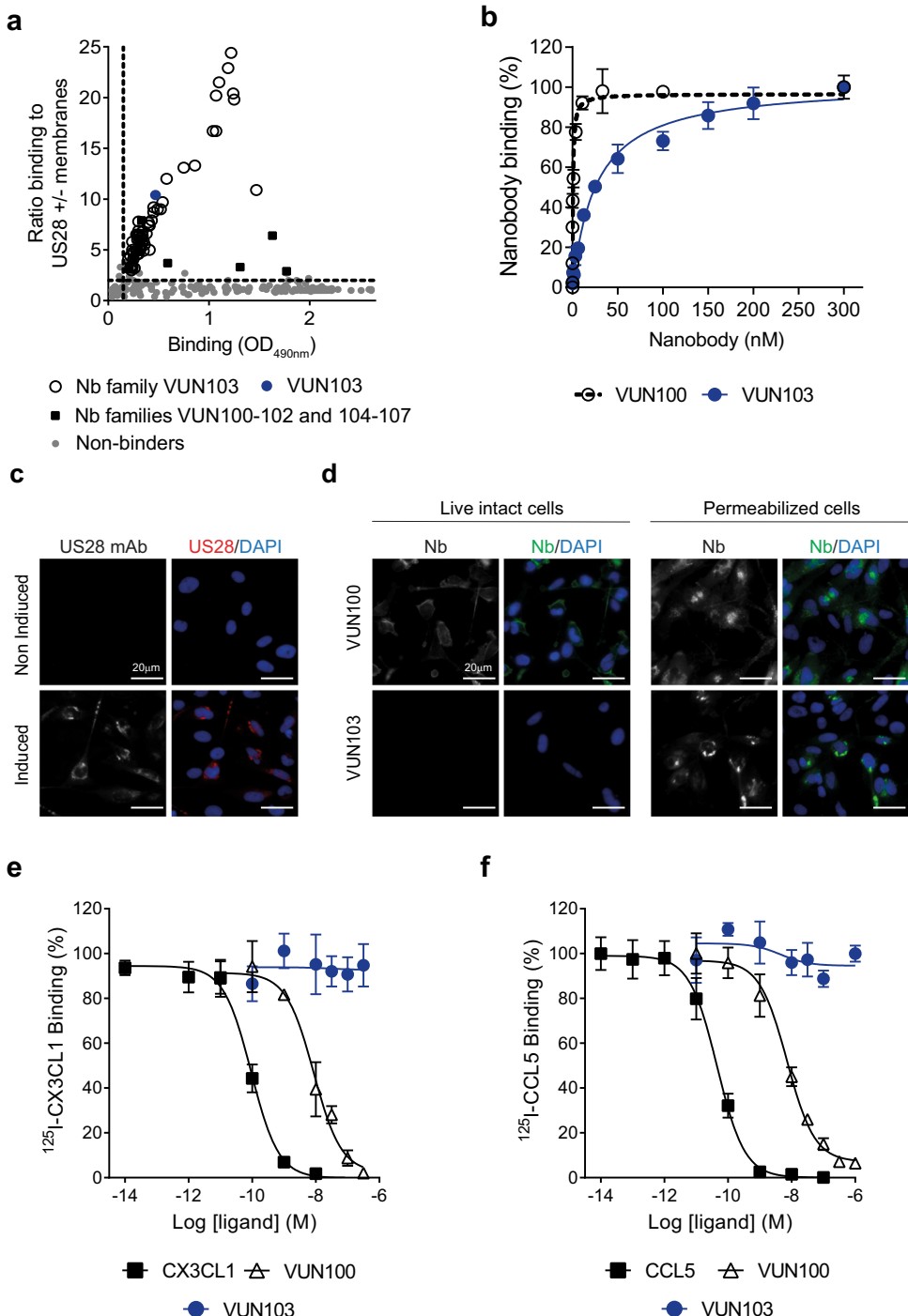

**Fig. 1 VUN103 recognizes an intracellular epitope of US28. a** Binding of nanobody-expressing phage, after selection on US28, to US28 measured by phage ELISA. The ratio of binding of phage to membranes with or without US28 was determined and plotted against the total binding of phages to US28-expressing membranes. Data represent one independent experiment. **b** Binding of the nanobodies VUN100 and VUN103 to US28-expressing membranes, as determined by ELISA. Binding was normalized to binding to US28-negative membranes to determine specific binding ($n = 3$ independent experiments). **c** Staining of US28 after 48 h of doxycycline-induction of US28-expressing U251 cells. Non-induced US28-expressing U251 cells served as negative control. Cells were fixed and permeabilized before US28 was visualized using a conventional anti-US28 antibody (US28 mAb; US28, red) and nuclei were stained with DAPI. Representative data of three independent experiments. **d** Binding of VUN100 and VUN103 to live, intact (non-permeabilized) and fixed and permeabilized US28-expressing U251 cells. For live cells, nanobodies were incubated with cells on ice before cells were fixed. For permeabilized cells, cells were fixed and permeabilized before incubation with nanobodies. Nanobody binding was detected using the Myc-tag and an anti-Myc antibody (Nb, green) and nuclei were stained with DAPI. Representative data of three independent experiments. **e, f** Displacement of $^{125}$I- CX3CL1 (**c**) and $^{125}$I-CCL5 (**d**) from different US28-expressing membranes by unlabeled ligand (CX3CL1, CCL5, VUN100 or VUN103) ($n = 3$ independent experiments). All are plotted as mean ± SD. Source data are provided as a Source Data file.

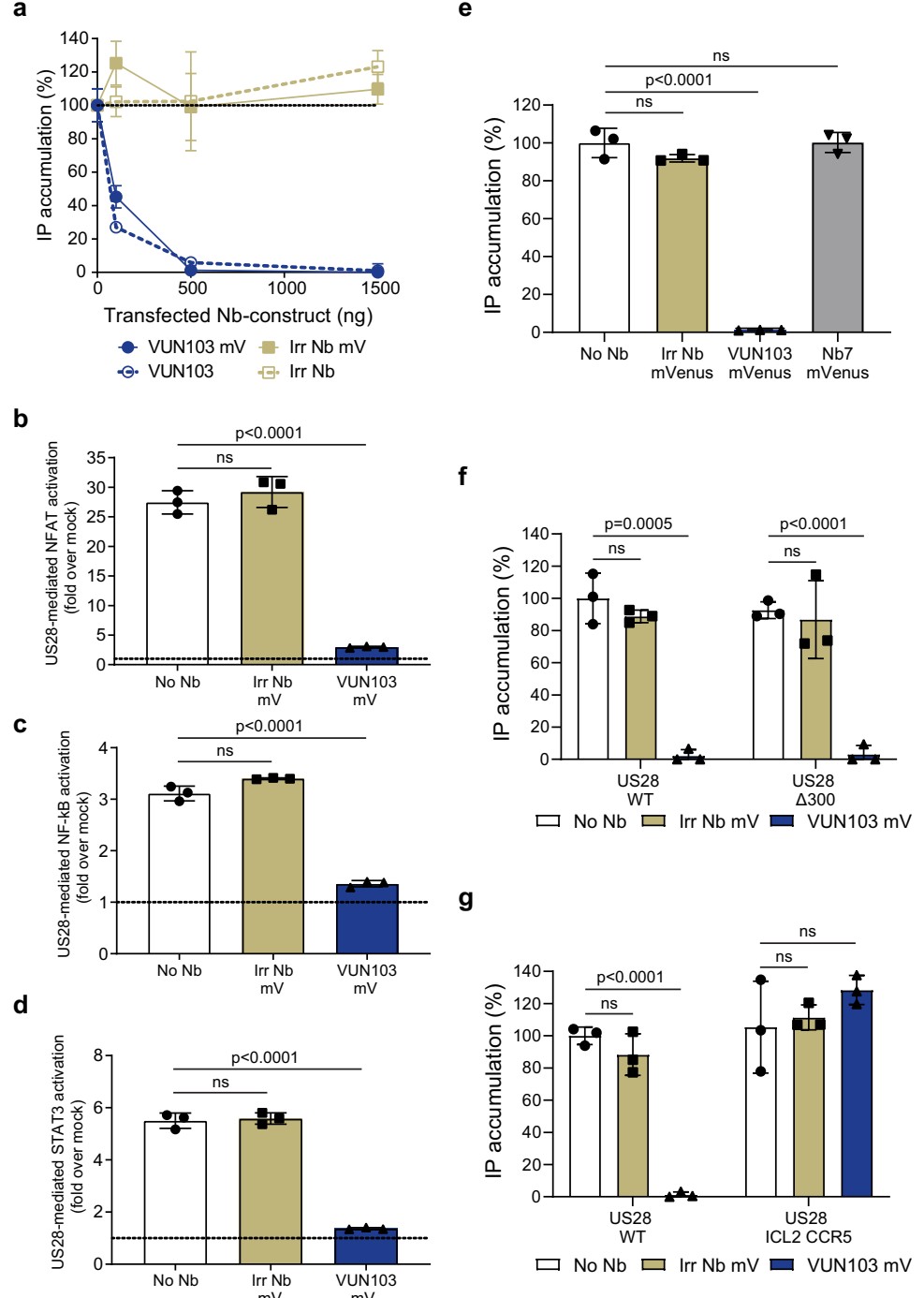

**Fig. 2 VUN103 inhibits US28 signaling by binding to the ICL2 loop. a** US28-mediated inositol phosphate (IP) accumulation of HEK293T cells expressing US28 and increasing amounts of VUN103, VUN103-mVenus (VUN103 mV), non-US28 targeting intrabody (Irr Nb) or non-US28 targeting intrabody-mVenus (Irr Nb mV) (*n* = 3 independent experiments). **b–d** Effect of US28 (No Nb) and US28 co-transfected with Irr Nb mV and VUN103 mV on NFAT activation (**b**), NF-κB activation (**c**) and STAT3 activation (**d**). Levels were normalized to cells transfected only with the reporter gene construct (*n* = 3 independent experiments). **e** US28-mediated IP accumulation of HEK293T cells expressing US28 (No Nb), US28 and Irr Nb mV, VUN103 mV, or Nb7-mVenus (Nb7 mV) (*n* = 3 independent experiments). **f**, **g** US28-mediated IP accumulation of HEK293T cells expressing HA-US28 wildtype (US28 WT), HA-US28 lacking a C-tail (US28 Δ300, **f** or an HA-US28 chimera with the ICL2 loop substituted with the corresponding ICL2 loop of CCR5 (US28 ICL2 CCR5, (**g**)). Cells expressed the US28 construct (No Nb) alone or in co-expressed with Irr Nb mV or VUN103 mV. All inositol phosphate accumulation levels were normalized to cells expressing US28 WT and no Nb-mVenus construct (*n* = 3 independent experiments). All data are plotted as mean ± SD. Statistical analyses were performed using unpaired two-tailed *t*-test. ns, *p* > 0.05. Source data are provided as a Source Data file.

While VUN103 blocked US28 signaling, Nb7 did not, indicating that VUN103 and Nb7 recognize distinct conformations of US28. To determine where VUN103 binds to US28, we used different US28 mutants, which either lacked the complete cytoplasmic tail (US28 Δ300, Fig. 2f) or where the ICL2 loop of US28 was replaced by those CCR5 (US28 ICL2 CCR5, Fig. 2g). Both US28 mutants showed similar levels of expression compared to wild-type US28 (Supplementary Fig. 4) and displayed similar levels of constitutive activity (Fig. 2f, g). While VUN103 inhibited the constitutive activity of US28 Δ300, it did not affect the constitutive activity of the US28 ICL2 chimera, indicating that ICL2 loop of US28 but not the cytoplasmic tail is important for binding of VUN103. In addition, we determined the binding of either VUN103 or Nb7 to US28 mutants of which the ICL loop 1 or 3 were replaced with the corresponding ICL loops of CCR5 (Supplementary Fig. 5). While swapping of the ICL1 loop did not affect the affinity of VUN103 for US28 ($24 \pm 7$ nM), binding of VUN103 to the US28 ICL3 CCR5-chimera was detected only at high concentrations. In line with the functional assays, no binding of VUN103 to the US28 ICL2 CCR5 chimera was seen. Likewise, we observed no binding of Nb7 to the US28 ICL2 CCR5 and US28 ICL3 CCR5 chimera, which is in line with previous crystallization studies[37]. Overall, these results indicated that both the ICL2 and ICL3 loop of US28 are important for the binding of VUN103 to US28.

We hypothesized that, by binding to the ICL2 and ICL3 loop of US28, VUN103 inhibits signaling by competing with the binding of G proteins. To assess this, we developed an US28-Bio ID2 fusion protein by linking the Bio ID2 protein to the cytoplasmic tail of US28 (Fig. 3a). Bio ID2 is a variant of biotin ligase that biotinylates lysines of any protein in close proximity and can thus be used to investigate protein-protein interactions[42]. First, receptor activity of US28 Bio ID2 was assessed (Fig. 3b). Despite similar expression levels, the US28 Bio ID2 fusion protein showed impaired (~30%) basal activity compared to US28 wild type. Nevertheless, it still displayed constitutive activity, which could be blocked by co-expression of the VUN103 intrabody. Addition of the Bio ID2 tag therefore did not affect the binding of VUN103 to the receptor. Next, we determined the biotinylation activity of the US28 Bio ID2 fusion protein (Fig. 3c). Mock transfected cells showed only a few clearly biotinylated polypeptides, corresponding to known endogenously biotinylated proteins[43,44]. Cells that express US28 Bio ID2 (with or without intrabody) showed a clear pattern of biotinylated proteins upon the addition of biotin. As expected, only cells that co-expressed US28 Bio ID2 and VUN103 intrabody showed a prominent biotinylated species around 45 kDa, which corresponds to the VUN103 intrabody (15 kDa) fused to the mVenus-tag (30 kDa) as also confirmed by MS analysis ($60.9 \pm 9.7$ VUN103-mVenus peptide counts (sequence coverage of 32%) vs $3.1 \pm 1.7$ non-US28 targeting Nb peptide counts (sequence coverage of 26.3%) as negative control).

We recovered biotinylated proteins, bound on streptavidin beads and assessed the presence of endogenously biotinylated pyruvate carboxylase (loading control), VUN103, and $G\alpha_q$ protein (Fig. 3d, e). While both intrabody constructs were detected in equivalent amounts in total cell lysates, only biotinylated VUN103 intrabody was recovered on streptavidin beads. Furthermore, while total levels of $G\alpha_q$ protein did not differ between the lysates, an increase in $G\alpha_q$ biotinylation was observed upon expression of US28 Bio ID2, reduced to background levels by co-expression of VUN103 (Fig. 3d, e). This was not seen for cells that co-express US28-Bio ID2 and the non-US28 targeting intrabody. We validated the immunoblotting results by determining the BRET between US28-Nluc and Venus-mini-$G\alpha_q$ construct upon co-expression with FLAG-tagged VUN103 intrabody or non-US28 targeting intrabody (Fig. 3f)[45].

Expression of the VUN103 intrabody resulted in significant lower BRET signals, indicating reduced interaction between US28-Nluc and the mVenus-mini-$G\alpha_q$. Despite higher expression levels, this reduction in BRET signal was not observed upon expression of the non-US28 targeting intrabody (Supplementary Fig. 6a).

Using a similar Bio ID2 setup, the effect of VUN103 on the constitutive coupling of β-arrestins to US28 was assessed. The expression of VUN103 resulted in significantly lower levels of biotinylated β-arrestin1/2 (Fig. 3g, h). This was not seen in cells that express the non-US28 targeting intrabody. To validate this observation, BRET between US28-Nluc and β-arrestin2-mVenus was determined upon expression of VUN103 or a non-US28 targeting intrabody (Fig. 3i and Supplementary Fig. 6b). Similar to $G\alpha_q$ protein, binding of VUN103 reduces the constitutive coupling of β-arrestin2 to US28. We conclude that VUN103 prevents constitutive coupling of both G proteins and β-arrestin1/2 to the constitutively active, ligand-unbound US28.

**VUN103 and Nb7 recognize distinct active states of US28.** Next, we examined the effect of the VUN103 intrabody on the binding of chemokines to US28. Membranes of HEK293T cells that express US28 with or without VUN103 or Nb7 were used to determine the affinity of CX3CL1 and CCL5 for US28 (Fig. 4a, b). Expression of VUN103 did not affect the binding affinities of CX3CL1 and CCL5 for US28 (Table 1). In contrast, a 10-fold increase in affinity of CX3CL1 (Fig. 4a) and CCL5 (Fig. 4b) was detected in the presence of Nb7 intrabody (Table 1). To further substantiate these effects, different nanobodies (non-US28 targeting nanobody (Irr Nb), VUN103 and Nb7) were genetically fused to the C-tail of US28, thereby providing a US28:nanobody ratio of 1:1 and maximally enabling the nanobodies to lock US28 in particular conformations[20]. Moreover, NanoLuc luciferase (NLuc) was fused to the N-terminus of US28 to allow measurement of binding affinities of fluorescently labeled chemokines (Fig. 4c, d). Fusion of Nb7 to US28 resulted in an approximate 7- and 9-fold increase in affinity of labeled CX3CL1 and CCL5, whereas no significant change was seen for VUN103 or the non-US28 targeting nanobody (Table 2). Moreover, fusion of Nb7 and VUN103 resulted in a significant increase in the maximal observed BRET ratios, consistent with a relative increase in accessible US28. This increase in accessible US28 upon intrabody binding could be a consequence of a reduction in constitutive internalization and degradation of the receptor, as might be caused by inhibition of β-arrestin1/2 recruitment (Fig. 4c, d, Table 2). Indeed, in HEK293T cells, expression of VUN103 and Nb7 significantly increased both total and surface expression of US28, as determined by ELISA (Fig. 4e). We conclude that Nb7 locks US28 in a ligand-bound-like active conformation that favors chemokine binding, whereas VUN103 recognizes an (apo)-conformation that is distinct from the ligand-bound, Nb7-stabilized US28 conformation. By binding to the intracellular side of the receptor, both intrabodies prevent the recruitment of G protein and β-arrestin, hereby blocking constitutive receptor signaling and internalization.

To determine whether VUN103 recognizes the apo-, constitutively active conformation of US28 as well as the ligand-bound conformation, we examined co-localization and recruitment of the different intrabodies to US28 in the absence and upon addition of CX3CL1 by confocal microscopy (Fig. 5a and Supplementary Fig. 7) on fixed cells. Cell surface HA-tagged US28 was labeled with fluorescent anti-HA antibodies. Co-localization of VUN103 and US28 was seen both in the absence and presence of CX3CL1. In contrast, although some co-localization of Nb7 and US28 was seen in the unstimulated cells, the extent of co-localization increased significantly upon the

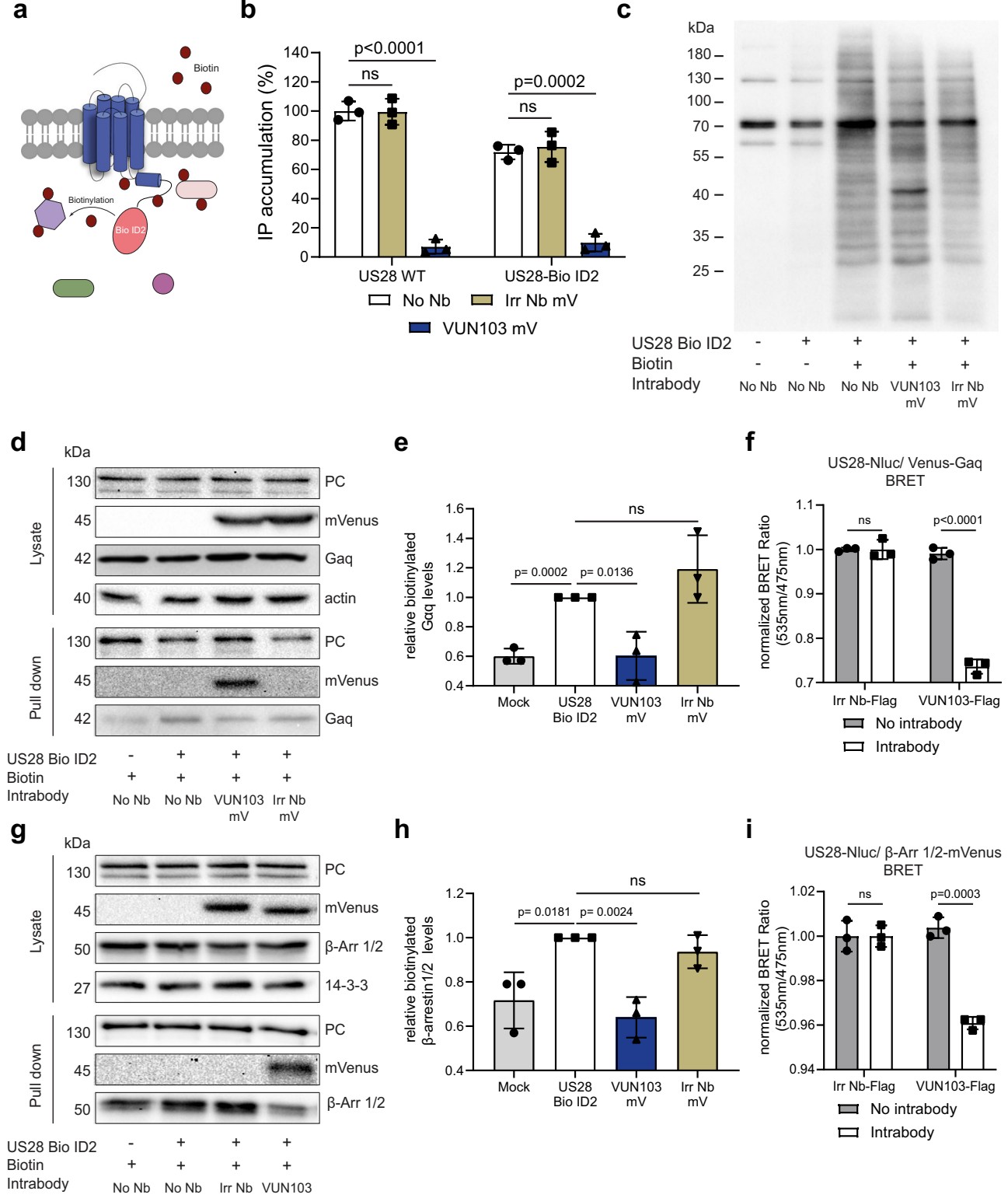

addition of CX3CL1. No co-localization of the non-US28 targeting intrabody and US28 was seen in the presence or absence of CX3CL1. To confirm the interaction of VUN103 or Nb7 with US28 in the absence or presence of chemokine binding, BRET between US28-Rluc and intrabody-mVenus was used. Close proximity of VUN103-mVenus or Nb7-mVenus with US28-Rluc was detected by saturation BRET (Fig. 5b). The addition of CX3CL1 slightly but significantly decreased the BRET

signal between US28-Rluc and VUN103-mVenus and increased the BRET between US28-Rluc and Nb7-mVenus (Fig. 5c). Similar results were obtained when the other US28 ligands CCL2, CCL3 and CCL5 were used (Supplementary Fig. 8). To confirm that these results were due to recruitment or competition of Gα_q, BRET between US28-Nluc and mVenus-mini-Gα_q upon co-expression of the different intrabodies, was assessed in the presence and absence of CX3CL1. In the absence of CX3CL1,

**Fig. 3 VUN103 competes with G$\alpha_q$ and β-arrestin1/2 binding. a** Overview of biotinylation of proximal proteins by US28-Bio ID2. **b** US28-mediated inositol phosphate (IP) accumulation in HEK293T cells expressing HA-US28 wildtype (US28 WT) or HA-US28-Bio ID2(No Nb) alone or co-expressed with non-US28 targeting intrabody-mVenus (Irr Nb mV) or VUN103-mVenus (VUN103 mV). IP levels were normalized to US28 WT No Nbt ($n = 3$ independent experiments). **c** Detection of biotinylated proteins in lysates from HEK293T cells with and without expression of HA-US28-Bio ID2 alone (No Nb) or withIrr Nb mV or VUN103 mV by Western blot using streptavidin-HRP. Representative blot shown from three independent biological replicates. **d** Detection of pyruvate carboxylase (PC), intrabody-mVenus (mVenus), G$\alpha_q$ and actin in lysates and biotinylated protein pulldown samples of HEK293T cells with or without HA-US28-Bio ID2 alone (No Nb) or withVUN103 mV or Irr Nb mV. Representative blot shown from three independent biological replicates. Samples are derived from the same experiment and blots were processed in parallel. **e** Quantification of biotinylated G$\alpha_q$ protein levels of biological triplicate samples. Data from three independent experiments using independent biological replicates. **f** BRET between US28-Nluc and Venus-mini-G$\alpha_q$ upon inducible expression of non-US28 targeting intrabody (Irr Nb-FLAG) VUN103 (VUN103-FLAG)(Intrabody). BRET signals were normalized to that in non-induced HEK293T cells (No intrabody) ($n = 3$ independent experiments). **g** Detection of PC, mVenus, β-arrestin1/2 (β−Arr 1/2), and 14-3-3 in lysates and biotinylated protein pulldown samples of HEK293T cells with or without HA-US28-Bio ID2 alone (No Nb) or with VUN103 mV or Irr Nb mV. Representative blot shown from three independent biological replicates. Samples are derived from the same experiment and blots were processed in parallel. **h** Quantification of biotinylated βArr-1/2 protein levels in biological triplicate samples. Data from three independent experiments using independent biological replicates. **i** BRET between US28-Nluc and β-arrestin2-mVenus upon inducible expression of Irr Nb-FLAG or VUN103-FLAG (Intrabody). BRET signals were normalized to that in non-induced HEK293T cells (No intrabody) ($n = 3$ independent experiments). All data are plotted as mean ± SD. Statistical analyses were performed using unpaired two-tailed $t$-test. ns, $p > 0.05$. Source data are provided as a Source Data file.

both VUN103 and Nb7 already reduced the BRET levels, compared to the non-US28 targeting intrabody (Supplementary Fig. 9a). However, even despite lower expression levels of VUN103, this decrease in BRET signal was significantly more pronounced for VUN103 compared to Nb7 (Supplementary Fig. 9b). Upon addition of CX3CL1, an increase of BRET signal between US28-Nluc and mVenus-mini-G$\alpha_q$ was observed in the presence of VUN103, while the opposite effect was seen for the Nb7 intrabody (Fig. 5d). This could indicate that CX3CL1 renders the US28 in a conformation that is less favorable for VUN103 but instead favors G protein binding, suggesting a competitive mode of binding.

Because the intrabodies compete with the binding of G$\alpha_q$ to US28 (Fig. 3), we asked whether we could modulate G protein competition with CX3CL1. The addition of CX3CL1 partially rescued VUN103-mediated inhibition of US28 signaling (Fig. 6a, left). This suggests that CX3CL1 allosterically reduces VUN103 binding and thereby prevents VUN103 from inhibiting US28-mediated signaling. In contrast, expression of Nb7 did not affect constitutive G$\alpha_q$-mediated signaling in the absence of CX3CL1. However, the addition of CX3CL1 resulted in decreased G$\alpha_q$ signaling. As these intrabodies showed opposing effects upon the addition of CX3CL1, we hypothesized that their ability to bind different conformations of US28 would allow them to completely inhibit US28 signaling when co-expressed. Indeed, co-expression of Nb7 with VUN103, and not the non-US28 targeting intrabody, resulted in full inhibition of US28 signaling, both in the absence or presence of CX3CL1 (Fig. 6a, right). The observed effects were not due to differences in expression levels (Fig. 6b, c). Both intrabodies thus recognize different US28 conformations with Nb7 favoring the ligand-bound-like conformation and VUN103 the apo ligand-unbound conformation.

**VUN103 inhibits US28-mediated oncogenic signaling and spheroid growth**. The observation that VUN103 binds the constitutively active conformation of US28 and blocks US28 signaling is of therapeutic interest. In glioblastoma, US28 signaling is associated with enhanced tumor growth. We were able to partially inhibit US28-mediated tumor growth using bivalent nanobodies that bind to the extracellular portion of US28[24]. To assess the therapeutic potential of VUN103 intrabody, we tested its application using a genetic approach. To that end, US28-expressing glioblastoma cells were transduced with non-US28 targeting intrabody-mVenus or VUN103-mVenus via lentiviral infection (Fig. 7a). The functionality of the intrabody was assessed by quantification of US28-enhanced spheroid growth (Fig. 7b, c). As

previously shown, US28 expression in U251 glioblastoma cells enhances spheroid growth as a result of its constitutive and oncomodulatory signaling. Moreover, we were able to inhibit spheroid growth up to 50% using the previously reported bivalent US28 nanobodies. Whereas the non-US28 targeting intrabody did not affect US28-enhanced spheroid growth, VUN103 significantly reduced US28-enhanced spheroid growth. We saw no differences in US28 levels for the different transduced cell lines, indicating that the effect was not due to differences in expression levels (Supplementary Fig. 10). Therefore, VUN103, upon lentiviral gene delivery, blocks US28 signaling in glioblastoma cells and can inhibit US28-mediated glioblastoma spheroid growth.

To confirm that this effect was mediated by binding of VUN103 to US28, US28-mediated accumulation of inositol phosphate was determined in U251 cells transiently expressing either US28 wildtype or the US28 ICL2 CCR5 in combination with either VUN103 or the non-US28 targeting intrabody (Supplementary Fig. 11a). At comparable expression levels of the different US28 and intrabody variants in the U251 cells (Supplementary Fig. 11b, c), VUN103 significantly inhibited the constitutive signaling of US28 wildtype and not US28 ICL2 CCR5. Inhibition of US28 signaling by expression of VUN103 is therefore mediated by binding of VUN103 to the intracellular loops of the receptor.

Finally, we assessed the effect of VUN103 in a more relevant infection setting. U251 glioblastoma cells were transduced with non-US28 targeting-intrabody-mVenus or VUN103-mVenus and were infected with wild-type HCMV (WT) or with HCMV lacking US28 (HCMV ΔUS28). VUN103 inhibited phosphorylation of STAT3, a known US28 signaling route, to similar levels as the uninfected U251 cells (Fig. 6d and Supplementary Fig. 12). In contrast, the expression of the non-US28 targeting intrabody did not affect STAT3 phosphorylation. To ensure proper quantification between different HCMV-infected samples, expression of HCMV-encoded immediate early proteins was checked and used to normalize phosphorylation levels (Fig. 6d and Supplementary Fig. 6). Both transduced cell lines expressed the intrabody at similar levels as evident from mVenus expression (Fig. 6d). Next, VEGF secretion of the different HCMV-infected cells was measured (Fig. 6e). We detected an increase in VEGF secretion upon infection with WT HCMV. Cells infected with HCMV ΔUS28, devoid of US28 expression, showed reduced VEGF secretion. In line with the STAT3 results, VUN103 reduced US28-mediated VEGF secretion to almost the same levels as HCMV ΔUS28 infected cells, while this was not seen for the non-US28 targeting intrabody.

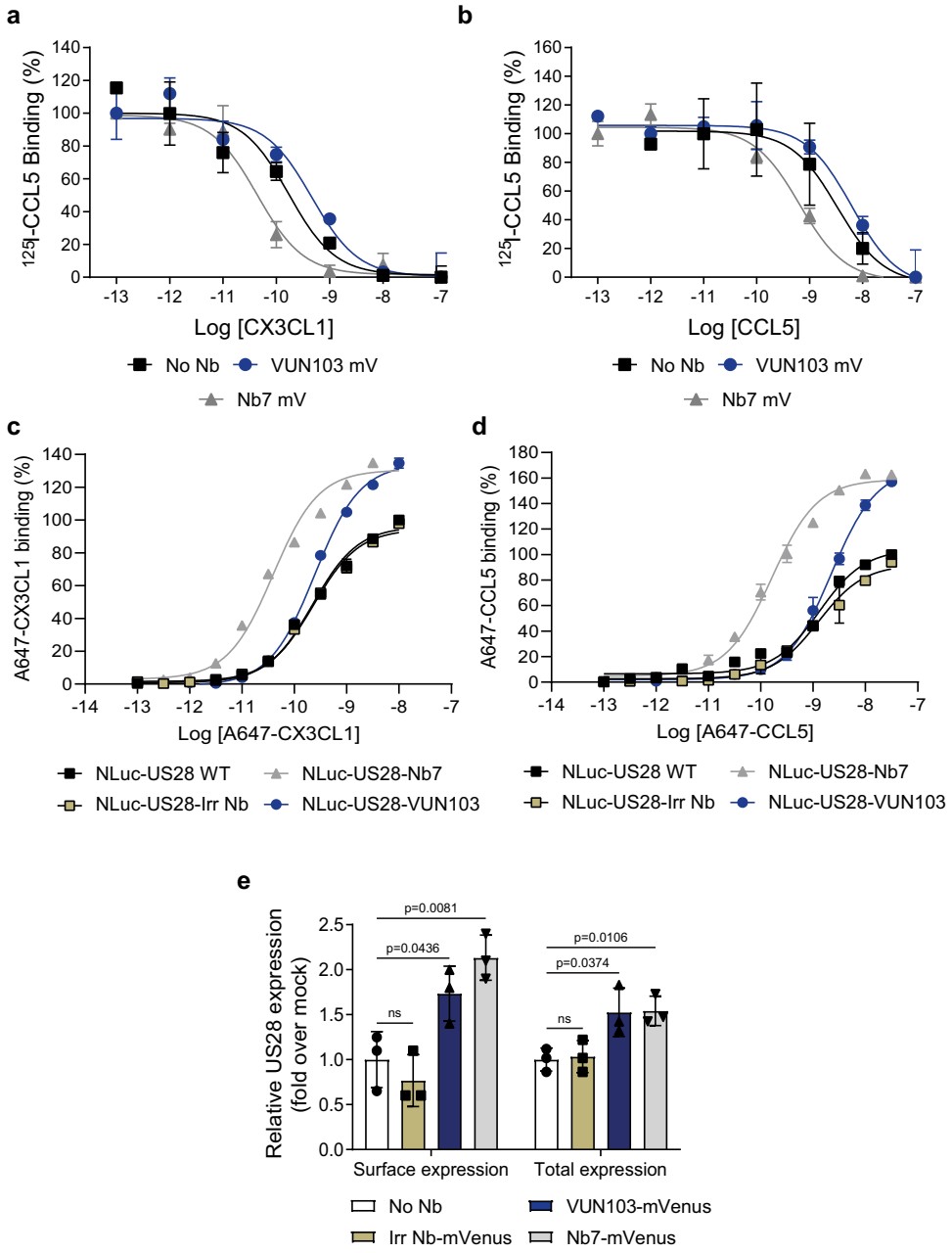

**Fig. 4 VUN103 and Nb7 differentially affect the affinity of US28 ligands. a–b** Displacement of $^{125}$I-CCL5 with unlabeled CX3CL1 (**a**) and unlabeled CCL5 (**b**) from different US28-expressing membranes co-expressing no intrabody (No Nb), VUN103-mVenus (VUN103 mV) or Nb7-mVenus (Nb7 mV) ($n = 3$ independent experiments). **c**, **d** BRET-based affinity determination of AlexaFluor (A)647-labeled CX3CL1 (**c**) and A647-labeled CCL5 (**d**) using membrane extracts of HEK293T cells expressing wildtype NLuc-US28 (NLuc-US28-WT) or NLuc-US28-nanobody fusions. In the case of the different NLuc-US28-nanobody fusions, a non-US28 targeting nanobody (NLuc-US28-Irr Nb), VUN103 (NLuc-US28-VUN103) or Nb7 (NLuc-US28-Nb7) was genetically fused to the C-terminus of NLuc-US28. Binding was determined after subtraction of the background (buffer only) and values were normalized to receptor expression. The percentage of binding was calculated by setting CX3CL1 or CCL5 binding at their highest concentrations to NLuc-US28-WT as 100% binding ($n = 3$ independent experiments). **e** ELISA measurements of surface and total US28 expression levels of HEK293T cells that express only HA-US28 or HA-US28 in combination with either a non-US28 targeting intrabody (Irr Nb)-mVenus, VUN103-mVenus or Nb7-mVenus. Surface US28 levels were determined on fixed cells. Total US28 was detected upon fixation and permeabilization. US28 was detected using its N-terminal HA tag and an anti-HA antibody. Expression levels were normalized to No Nb condition ($n = 3$ independent experiments). All data are plotted as mean ± SD. Statistical analyses were performed using unpaired two-tailed $t$-test. ns, $p > 0.05$. Source data are provided as a Source Data file.

## Discussion

We report the characterization of a nanobody (VUN103) that binds intracellularly to a constitutively active apo (ligand-free) conformation of the chemokine receptor US28. A comparison of the effect of VUN103 and Nb7 (a nanobody recognizing the ligand-bound conformation) on chemokine binding, US28 signaling, and

localization, shows that the constitutively active (ligand-free) conformation of US28 is distinct from the ligand-bound, active conformation.

GPCRs are highly dynamic proteins that can adopt different conformations[4]. Although the initial model postulated an equilibrium between the inactive and active conformation of a

receptor, this two-state model has been replaced by a model that proposes multiple active and inactive conformations[3]. Nanobodies are ideal tools to stabilize and explore these different conformations, as shown for the β1-AR, β2-AR, Muscarinic 2 receptor, μ-opioid and κ-opioid receptor[14–17,37,46–51]. Nanobodies that stabilize an active GPCR conformation interact with the G protein interface and act as G protein mimetics, thereby inhibiting G protein-mediated signaling. The ICL2 and ICL3 loops are important interfaces between Gα protein and the GPCR[14,52,53]. By using US28-ICL chimeric receptor constructs, we confirmed the importance of the interaction of VUN103 and Nb7 with the ICL2 and ICL3 loop of US28. These interactions were also apparent in the crystal structure of CX3CL1-bound US28 and Nb7[37]. Hence, these loops are crucial for the recognition and/or stabilization of a US28-conformation in a complex with a nanobody. In addition, we detected reduced β-arrestin coupling and increased surface and total expression levels of US28 upon co-expression of VUN103 (and Nb7). Similar increases were observed in our BRET-based binding experiments using US28-nanobody fusions. These results suggest that the binding of VUN103 to US28 influences receptor internalization and degradation, possibly via inhibition of constitutive β-arrestin-coupling. However, we cannot exclude that VUN103 competes with other US28-interacting proteins involved in receptor internalization, such as AP-2 and Dynamin. For instance, previous studies have shown that US28 internalizes via β-arrestin-independent pathways (clathrin-mediated and caveolae-mediated internalization)[54,55].

Although both the Nb7 and VUN103 nanobodies bind to the same intracellular loops, the functional consequences of their binding on US28 signaling are different. While VUN103 blocks constitutive signaling by competition with G proteins, Nb7 does not affect constitutive US28 signaling. This is interesting because of the ability of VUN103 to bind the apo-conformation and its ability to compete for G protein binding[38], as confirmed also by our BRET experiments. However, the lack of effect of Nb7 on constitutive US28 signaling could be due to the lower affinity of Nb7 to apo-US28. In contrast, the binding of CX3CL1 and other US28 ligands results in increased binding of Nb7 and a significant decrease of VUN103 binding to US28. The results of these functional assays were corroborated by confocal microscopy and BRET experiments. Nb7 and VUN103 recognize different conformational states of US28 (Fig. 8). The binding of CX3CL1 induces subtle but significant changes in the binding of both nanobodies to US28. This suggests that the apo-conformation and ligand-bound conformation of US28 differ only slightly. This would not be all that surprising, given that US28 is characterized by high constitutive signaling activity, in line with previous studies where ligand binding only marginally superactivated US28 signaling. US28 may thus have a more modest dynamic range[24,38].

Nanobodies that stabilize the active and inactive conformation of the β2-AR are able to increase or decrease the affinity of β2-AR ligands for the receptor[47]. In our ligand competition and BRET-based ligand binding assays using US28-nanobody fusion constructs, the expression of VUN103 did not affect the affinities of CX3CL1 and CCL5. In contrast, the expression of Nb7 induced a marked increase in affinity for both ligands. We therefore conclude that VUN103 does not stabilize the inactive conformation of US28. While VUN103 and Nb7 are interesting pharmacological tool compounds to distinguish and define different conformations of US28, one could envision other uses for these nanobodies. GPCRs signal not only from the plasma membrane but also from endosomes and the Golgi apparatus[18,19,56]. Both Nb7 and VUN103 can be used to investigate spatiotemporal signaling and the presence of different conformations of a constitutively active receptor in different cell compartments. However, it would be important to regulate and optimize the expression levels of these intrabody biosensors to minimize competition with US28-interacting proteins, thereby influencing (subcellular) receptor levels and/or signaling. Moreover, VUN103 and Nb7 could serve as screening tools to find new small molecules that target the G protein interface of US28. Recent crystal structures of chemokine receptors have sparked renewed interest in targeting a conserved G protein-GPCR interface[57]. US28 is important in different disease manifestations. To date, no potent and selective small molecules have been found that target and inhibit this receptor[39,58]. In contrast to known US28 inhibitors, VUN103 blocked US28 signaling completely. BRET-based screening, using VUN103 (and Nb7) might help identify conformation-specific small molecules that act as (allosteric) antagonists or inverse agonists.

Finally, VUN103, when expressed intracellularly, may have therapeutic potential as indicated from our 3D spheroid model and from experiments performed in HCMV-infected cells. Our previously described bivalent US28 nanobodies inhibit the constitutive activity by only 50%[24]. Full inhibition of US28 signaling could provide more potent therapeutic strategy to treat HCMV-positive tumors. Interestingly, co-expression of VUN103 and Nb7 could have beneficial inhibitory effects as shown in our inositol phosphate accumulation assays. Gene therapy vectors, such as adeno-associated viruses, allow long-term expression of therapeutics in vivo[59,60]. As an example, a bispecific nanobody that inhibits degradation of gelsolin was introduced in vivo using a gene therapy vector in a preclinical model, making a similar strategy plausible for therapeutic use of VUN103[61].

**Table 1 pKi values of CX3CL1 and CCL5 to different membrane extracts.**

| Membrane extract | pKi CX3CL1 (Mean + SD) | pKi CCL5 (Mean + SD) |
|---|---|---|
| US28 WT | 9.9 ± 0.2 | 8.8 ± 0.1 |
| US28 WT + VUN103-mVenus | 9.7 ± 0.3 | 8.8 ± 0.2 |
| US28 WT + Nb7-mVenus | 10.7 ± 0.04 | 9.8 ± 0.1 |

**Table 2 pK$_D$ values and relative accessible US28 of A647-labeled CX3CL1 to different NLuc-US28-nanobody fusion constructs.**

| US28 construct | pK$_D$ A647-CX3CL1 (Mean + SD) | Relative accessible US28 A647-CX3CL1 (Mean + SD) | pK$_D$ A647-CCL5 (Mean + SD) | Relative accessible US28 A647-CCL5 (Mean + SD) |
|---|---|---|---|---|
| NLuc-US28 WT | 9.6 ± 0.1 | 1 ± 0.0 | 8.8 ± 0.06 | 1 ± 0.0 |
| NLuc-US28-Irr Nb | 9.6 ± 0.1 | 0.96 ± 0.10 | 8.8 ± 0.1 | 1.02 ± 0.10 |
| NLuc-US28-VUN103 | 9.6 ± 0.06 | 1.30 ± 0.07 | 8.8 ± 0.1 | 1.45 ± 0.10 |
| NLuc-US28-Nb7 | 10.4 ± 0.02 | 1.37 ± 0.03 | 9.8 ± 0.1 | 1.61 ± 0.04 |

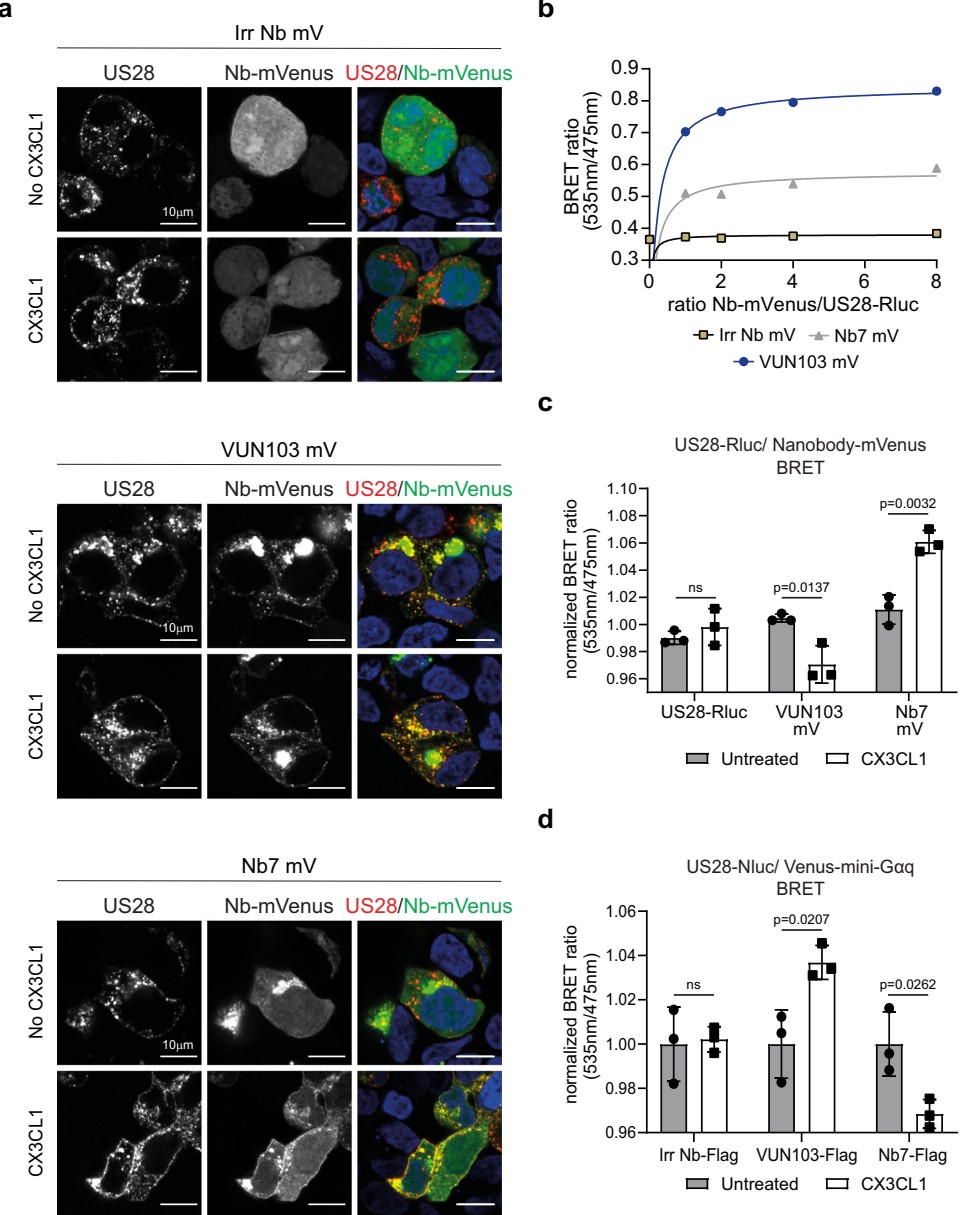

**Fig. 5 Effect of CX3CL1 on intrabody binding to US28. a** Confocal microscopy of HEK293T cells expressing HA-US28 and non-US28 targeting intrabody-mVenus (Irr Nb mV), VUN103-mVenus (VUN103 mV) or Nb7-mVenus (Nb7 mV) with (CX3CL1) or without (No CX3CL1) the addition of 30 nM CX3CL1. US28 was stained with an anti-HA Alexa Fluor555 antibody on ice before the addition of CX3CL1 to stain the US28 receptor population on the surface. Cells were fixated 20 min after the addition of CX3CL1 and stained US28 (US28 and different nanobody-mVenus (Nb-mVenus) constructs were visualized. Representative data of two independent experiments. **b** Saturation BRET with HEK293T cells expressing US28-Renilla luciferase (US28-Rluc) and different ratios of Irr Nb mV, VUN103 mV, or Nb7 mV. Data were plotted as the ratio of nanobody-mVenus constructs (Nb-mVenus) and US28-Rluc ($n = 3$ independent experiments). **c** BRET using HEK293T cells expressing US28-Rluc, US28-Rluc, and VUN103 mV or Nb7 mV 30 min after addition of 30 nM CX3CL1 (CX3CL1). BRET signal was normalized to the signal of unstimulated (untreated) HEK293T cells expressing US28-Rluc only at timepoint $t = 0$ ($n = 3$ independent experiments). **d** BRET with HEK293T cells expressing US28-Nluc, Venus-mini-G$\alpha_q$, and an inducible FLAG-tagged non-US28 targeting intrabody (Irr Nb-FLAG), FLAG-tagged VUN103 (VUN103-FLAG) or FLAG-tagged Nb7 (Nb7-FLAG) 30 min after addition of 100 nM CX3CL1 (CX3CL1). Expression of FLAG-tagged intrabodies was induced using 100 nM of tebufenozide, one day prior to read-out. BRET signal was normalized to the signal of the unstimulated (Untreated) HEK293T cells ($n = 3$ independent experiments). All data are plotted as mean ± SD. Statistical analyses were performed using an unpaired two-tailed $t$-test. ns, $p > 0.05$. Source data are provided as a Source Data file.

In conclusion, the constitutively active state of the viral chemokine receptor US28 differs from the ligand-bound conformation, as shown using two distinct nanobodies. Differences between conformational states of ligand-bound and free GPCRs may well apply to other constitutively active receptors and provide a means to selectively target the constitutive activity of such GPCRs. VUN103 and Nb7 provide the specific means to selectively discern functional consequences of ligand-independent and dependent signaling. Moreover, they are ideal tools to identify other therapeutic compounds that target the G protein-GPCR interface.

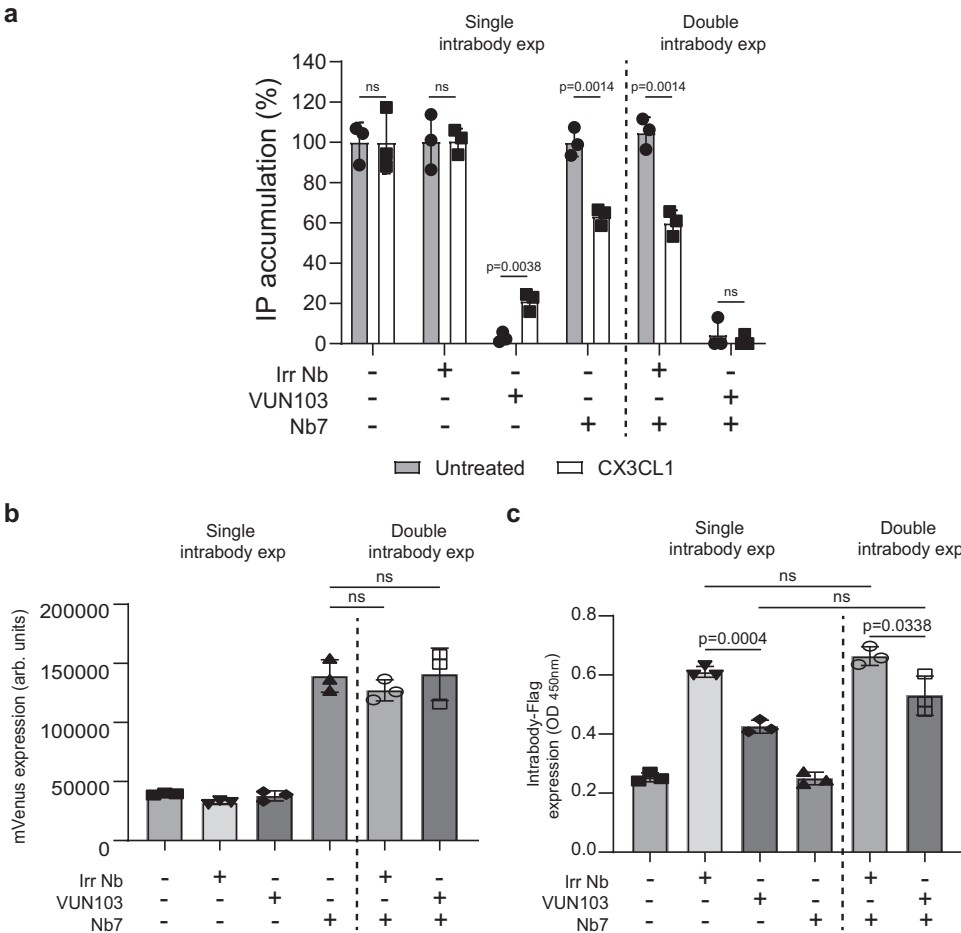

**Fig. 6 VUN103 and Nb7 have opposing effect on constitutive and ligand-induced signaling. a** Effect of non-US28 targeting FLAG-tagged intrabody (Irr Nb), FLAG-tagged VUN103 (VUN103), and/or Nb7-mVenus (Nb7) on US28-mediated accumulation of inositol phosphate (IP) in US28-expressing HEK293T cells that were untreated or stimulated with 30 nM CX3CL1 (CX3CL1). Expression of FLAG-tagged intrabodies was induced using 100 nM of tebufenozide, one day prior to read-out. Cells expressing a single intrabody (Single intrabody exp) or two intrabodies (Double intrabody exp) are divided by a dashed line. Accumulation of inositol phosphates were normalized to untreated US28-expressing cells ($n = 3$ independent experiments). **b** Determination of Nb7-mVenus expression levels by measuring the fluorescence intensity of mVenus. Cells expressing a single intrabody (Single intrabody exp) or two intrabodies (Double intrabody exp) are divided by a dashed line ($n = 3$ independent experiments). **c** Expression of FLAG-tagged intrabody was determined via ELISA by the C-terminal FLAG-tag and an anti-FLAG antibody. Cells expressing a single intrabody (Single intrabody exp) or two intrabodies (Double intrabody exp) are divided by a dashed line ($n =$ three independent experiments). All data are plotted as mean ± SD. Statistical analyses were performed using an unpaired two-tailed $t$-test. ns, $p > 0.05$. Source data are provided as a Source Data file.

## Methods

**DNA constructs**. The pVUN014 phagemid vector was a gift from Prof. Dr. H.J. de Haard (Argen x BV, Zwijnaarde, Belgium). The pET28a vector for periplasmic production of nanobodies in E.coli was described previously[62]. The pcDEF3 vector was a gift from Langer[63]. The pLenti6.3/To/V5-DEST for lentiviral transduction was described previously[64]. The pEUI(+) vector was a gift from Hyunju Ro (Addgene plasmid #111834). Genes encoding US28 mutants US28 Δ300 (removal of last 54 amino acids), US28 ICL2 CCR5 (swapping of ICL2 of US28 with the ICL2 of CCR5), mVenus and US28 Bio ID2 were ordered from Eurofins (Ebersberg, Germany) and subcloned to the pcDEF3 vector. US28 ICL1-CCR5 and ICL3-CCR5 (swapping of ICL1 or ICL3 of US28 with the respective ICL of CCR5), and the previously described NES-Venus-mGsq71 (mini-Gαq) construct were ordered from Twist Bioscience (San Francisco, California, United States)[45] and subcloned to the pcDEF3 vector. To generate the intrabody and intrabody-mVenus constructs, the PCR-amplified nanobodies and mVenus were recloned in the pcDEF3 vector. The pcDEF3 vectors encoding the US28-Nb fusion mutants were generated by fusing the PCR-amplified nanobodies with an N-terminal 10GS linker at the C-terminus of US28. US28 constructs with N-terminal NLuc were generated by replacing CXCR4 with US28 in the NLuc-CXCR4 pcDNA3.1 (+) construct, which was described previously[65], and subcloning it to the pcDEF3 vector. FLAG-tagged intrabodies were generated by subcloning nanobodies with a C-terminal FLAG tag to the pEUI(+) vector containing the ecdysone-inducible expression system[66]. Plasmids for lentiviral transduction were generated by subcloning US28 ICL2 CCR5 and intrabody constructs from the pcDEF3 vector to the pLenti6.3/To/V5-DEST vector. The pcDEF3 vector containing the Histamine 1 receptor construct

has been described previously[67]. An overview of all primers used for cloning can be found in Supplementary Table 1.

**Cell culture**. HEK293T cells were purchased from ATCC (Wesel, Germany). Doxycycline-inducible US28 expressing HEK293T cells (HEK293T-iUS28) and U251 cells (U251-iUS28) were described previously[24]. Inducible intrabody and U28 ICL2 CCR5 cell lines were generated by lentiviral transduction of U251 and/or U251-iUS28 cells. Lentiviruses were produced by transfection of HEK293T cells with packaging vectors pMD2.G and psPAX2 (a gift from Didier Trono (Addgene plasmid 12260)) and US28 ICL2 CCR5 or intrabody-mVenus pLenti6.3/To/V5-DEST plasmids. To induce US28 and intrabody-mVenus expression, cells were stimulated with doxycycline (1 μg/ml, D9891, Sigma-Aldrich, Saint Louis, Missouri, USA) for 24–48 h. To induce intrabody-FLAG expression, cells were stimulated with tebufenozide (100 nM, Sigma-Aldrich). Cells were grown at 5% $CO_2$ and 37 °C in Dulbecco's Modified Eagle's Medium (Thermo Fisher Scientific, Waltham, Massachusetts, USA) supplemented with 1% Penicillin/Streptomycin (Thermo Fisher Scientific) and 10% Fetal Bovine Serum (FBS, Thermo Fisher Scientific). In the case of U251 cells, FBS was heat inactivated for 20 min at 60 °C. HCMV merlin wildtype and HCMV ΔUS28 virus have been described previously[68].

**Transfection of adherent cells**. Two million HEK293T cells were plated in a 10 cm[2] dish (Greiner Bio-one, Kremsmunster, Austria). The next day, cells were transfected with a total of 5 μg DNA and 30 μg 25 kDa linear polyethyleneimine

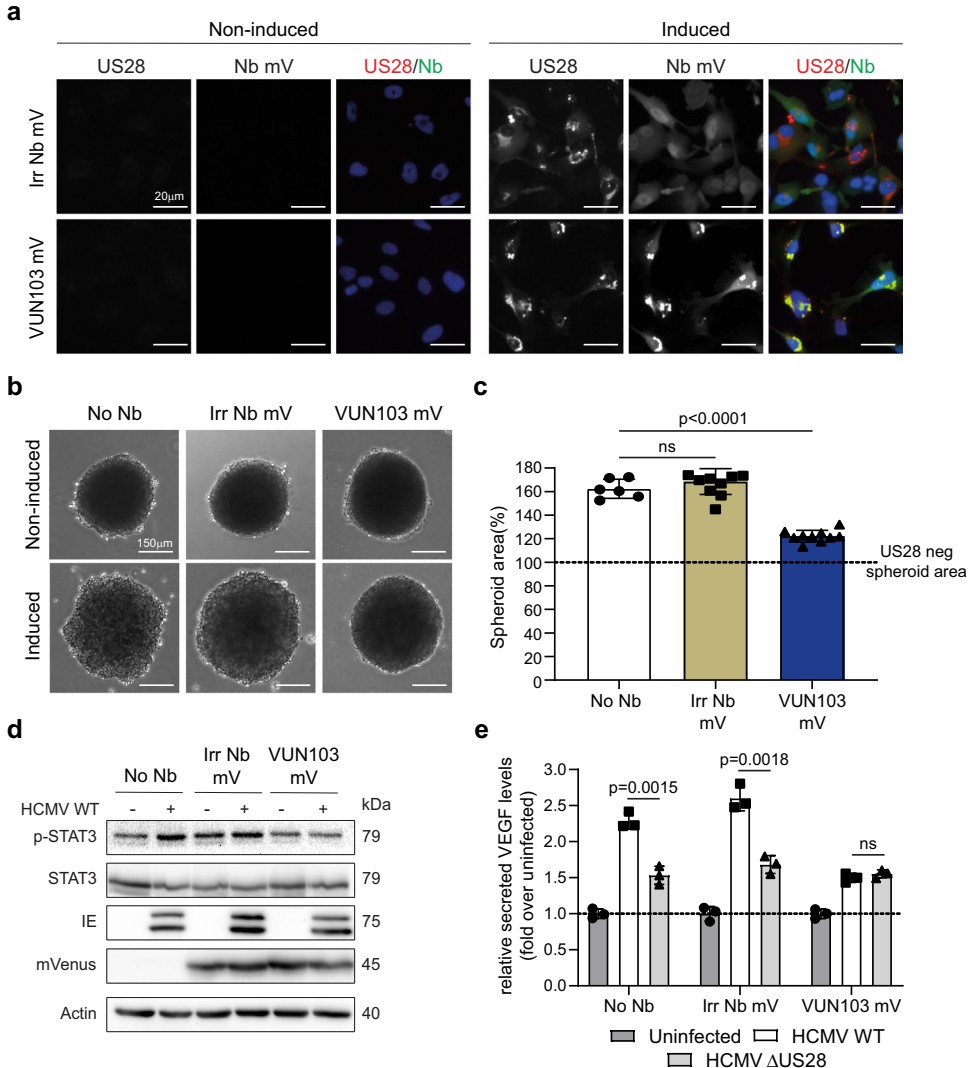

**Fig. 7 VUN103 inhibits US28-mediated glioblastoma spheroid growth. a** U251 with inducible US28 expression (No Nb) were transduced with lentiviral vector expressing inducible non-US28 targeting intrabody-mVenus (Irr Nb mV) or VUN103-mVenus (VUN103 mV). Cells were (induced) or were not (Non-induced) induced with doxycycline. US28 was detected using an anti-US28 antibody (US28, red). Nanobody binding was detected using the mVenus-tag (Nb mV, green). Representative data of three independent experiments. **b** Cells were seeded in hanging droplet plates and expression of US28 (and intrabodies) were or were not induced with doxycycline. Representative data of three independent experiments. **c** Spheroid areas were quantified and normalized to the spheroid area of non-induced US28 negative (US28 neg) spheroid (*n* = 6 spheroids for non-induced and induced No Nb samples; *n* = 8 spheroids for non-induced Irr Nb mV samples; *n* = 12 spheroids for induced Irr Nb mV samples; *n* = 7 spheroids for non-induced VUN103 mV samples; *n* = 11 spheroids for induced VUN103 mV samples). **d** Western blot of phospho-STAT3 (p-STAT3), total STAT3, immediate early proteins (IE), intrabody-mVenus (mVenus) and actin of U251 cells (No Nb), U251cells transduced with a lentiviral vector expressing inducible non-US28 targeting intrabody-mVenus (Irr Nb mV) or VUN103-mVenus (VUN103 mV). Cells were not infected or infected with HCMV wild-type virus (WT). Representative blot shown from three independent experiments using independent biological replicates. Samples are derived from the same experiment and blots were processed in parallel. **e** U251 cells (No Nb), U251cells transduced with a lentiviral vector expressing inducible non-US28 targeting intrabody-mVenus (Irr Nb mV) or VUN103-mVenus (VUN103 mV)were uninfected or infected with HCMV wild-type virus (WT) or HCMV virus lacking US28 (HCMV ΔUS28). Conditioned serum of infected cells was harvested and VEGF levels were determined using ELISA (*n* = 3 independent experiments). All data are plotted as mean ± SD. Statistical analyses were performed using an unpaired two-tailed *t*-test. ns, *p* > 0.05. Source data are provided as a Source Data file.

(PEI; Sigma-Aldrich) in 150 mM NaCl solution. The DNA-PEI mixture was vortexed for 10 seconds and incubated for 15 min at room temperature (RT). Subsequently, the mixture was added dropwise to the adherent HEK293T cells.

**Cell suspension transfection**. HEK293T Cells were transfected with a total of 1 μg DNA and 6 μg 25 kDa linear PEI (Sigma-Aldrich) in 150 mM NaCl solution per 1 million cells. The DNA-PEI mixture was vortexed for 10 seconds and incubated for 15 min at room temperature (RT). HEK293T cells were detached with Trypsin (Gibco, Thermo Fisher Scientific) and resuspended in DMEM (Thermo Fischer). The HEK293T cell suspension was added to DNA-PEI mixture and cells were seeded.

**Membrane extract preparation**. HEK293T-iUS28 or U251-iUS28 were induced with doxycycline as described above. In the case of transfected cells, HEK293T cells were transfected as described above and protein expression was allowed to proceed for 24–48 h. Media was then removed and cells were washed once with cold PBS. Next, cells were detached and resuspended in cold PBS. Cells were centrifuged at 1500 × *g* at 4 °C, resuspended in cold PBS, and again centrifuged at 1500 × *g* at 4 °C. The pellet was resuspended in membrane buffer (15 mM Tris-Cl, 0.3 mM EDTA, 2 mM MgCl2, pH 7.5) and disrupted by the homogenizer Potter-Elvehjem at 1200 rpm.

**Phage production**. *E.coli* TG1 bacteria containing either immune VHH libraries (round 1) or enriched VHH libraries (round 2 or 3) in phagemid vector pVUN014

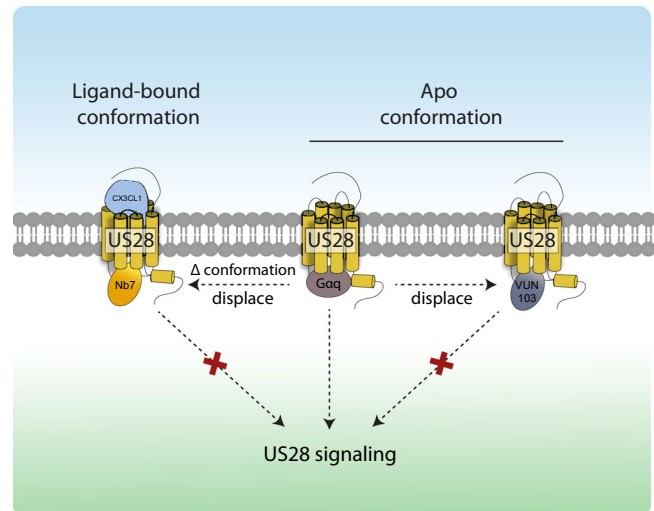

**Fig. 8 VUN103 and Nb7 recognize different US28 conformations.** Apo-US28 constitutively binds Gαq proteins resulting in constitutive signaling. VUN103 recognizes the US28 apo conformation and displaces Gαq proteins resulting in inhibition of US28 signaling. CX3CL1 binding to US28 results in a conformational change (Δ conformation) resulting in superactivation of US28 receptor. The ligand-bound US28-conformation is recognized by Nb7, resulting in competition with Gαq proteins, and subsequent inhibition of US28 signaling.

were inoculated in 2xTY broth containing 100 µg/mL ampicillin (Melford Biolabs ltd., Ipswich, UK) and 2% (w/v) glucose and grown until OD600 of 0.5. For the first round of selection, the two previously described phagemid libraries[39] were pooled. The amount of inoculate exceeded approximately >10 times the estimated library diversity (i.e., number of transformants). Cultures were infected with VCSM13 helper phage (Stratagene, San Diego, California, USA) at a phage-bacteria ratio of 10:1–20:1. Cultures were grown for 30 min at 37 °C while stationary, followed by 30 min at 37 °C while shaking. Bacteria were centrifuged at 4300 g, and the pellet was resuspended in 2xTY containing 50 µg/mL kanamycin (Melford Biolabs Ltd.) and 100 µg/mL ampicillin. The bacteria were grown overnight at 28 °C to allow phage production. The next day, the culture was centrifuged at 4300 g, and the supernatant was added to ice-cold 20% PEG6000/2.5 M NaCl (ratio 4:1) and incubated for 30 min on ice. The supernatant was centrifuged at $3500 \times g$, and the phage pellet was resuspended in PBS. The phage suspension was centrifuged, and the supernatant was again added to ice-cold 20% PEG6000/2.5 M NaCl and incubated for 10 min on ice. The supernatant was centrifuged again, and the phage pellet was resuspended in PBS.

**Phage display selections.** Three rounds of phage selections were performed using membrane extracts of the inducible US28 (VHL/E strain) HEK293T or U251 cell lines. Fifty µg of membrane extracts was coated in a 96-well MicroWell MaxiSorp flat bottom plate (Sigma-Aldrich) overnight at 4 °C. Wells were washed three times with PBS and blocked with 2% (w/v) skim milk (Sigma-Aldrich) in PBS for 1 h at RT. Phage were diluted tenfold (round 1) or hundredfold (rounds 2 and 3) in 0.2% (w/v) skim milk in PBS and incubated in the wells for 2 h at RT. Phage were first incubated with 250 µg of U251 membrane extracts for 1 h head-over-head (20 rpm) at RT in case a counter-selection was performed during round 3. The membrane extract-phage mix was centrifuged at $3500 \times g$, and the supernatant was added to wells containing the membrane extracts of the induced US28 U251 cell line. After incubation with phages, wells were washed 20 times with PBS, with every fifth washing step followed by an incubation step of 10 min on a shaker. Remaining phage were eluted with 10 mg/mL trypsin (Sigma-Aldrich) for 30 min at RT, and the eluate was mixed with 4 mg/mL 4-benzenesulfonyl fluoride hydrochloride (Sigma-Aldrich). Eluted phages were rescued by infecting TG1 cells (OD600 of 0.5) and then grown overnight at 37 °C. Rescued phages were used for subsequent rounds of phage display. After 2 and 3 rounds of selection, bacteria were plated, and single colonies were grown in a 96-well plate containing 2xTY broth and 100 µg/mL ampicillin.

**Phage ELISA.** After 2 and 3 rounds of selections, single clones were picked and grown in 2xTY 100 µg/mL ampicillin at 37 °C. When the cultures reached an OD600 of 0.5, the bacteria were infected with VCSM13 helper phage (final concentration $3.75 \times 10^{13}$ pfu/mL). Cultures were grown for 30 min without shaking, followed by 30 min at 37 °C while shaking. 2xTY and 100 µg/mL ampicillin and kanamycin were added to obtain a final concentration of 50 µg/mL kanamycin.

Cultures were grown overnight at 28 °C. Twenty-five µg of membrane extracts with or without US28 was coated in a 96-well MicroWell MaxiSorp flat bottom plate (Sigma-Aldrich) overnight at 4 °C. The next day, wells were washed three times with PBS and blocked with 3% (w/v) skim milk in PBS for 1 h at RT. Phage cultures were centrifuged at $3500 \times g$. The phage-containing supernatant was added to 3% (w/v) skim milk in a 1:1 ratio and incubated for 1 h at RT. Blocked phage solution was added to the MicroWell MaxiSorp flat bottom plate containing membrane extracts with and without US28. Phages were incubated for 2 h. at RT. Wells were then washed five times with PBS. Mouse-anti-M13-HRP (GE-Healthcare, Chicago, Illinois, USA) was diluted 1:5000 in 3% (w/v) skim milk in PBS and incubated for 1 h at RT. The plates were washed five times with PBS. O-Phenylenediamine (OPD) solution (2 mM OPD; Sigma-Aldrich, 35 mM citric acid, 66 mM Na2HPO4, 0.015% H2O2, pH 5.6) was added to the wells, and the reaction was stopped with 1 M H2SO4. Absorbance was measured at 490 nm with a PowerWave plate reader (BioTek, Winooski, Vermont, USA). Positive hits were sent for sequencing (LGC Genomics, Berlin, Germany) and grouped in different families based on the CDR3 sequence.

**Nanobody production.** Nanobody genes were recloned in frame with a Myc-His6 tag in a pET28a production vector and were transformed into BL21 + E. coli by heat shock. Transformed BL21 + E. coli were grown in an orbital shaker at 37 °C in Terrific Broth containing 50 µg/mL kanamycin. When the culture reached an OD600 of 0.5, nanobody production was induced by the addition of isopropyl-β-D-thiogalactopyranoside (Sigma-Aldrich) to a final concentration of 1 mM. Incubation then continued at 37 °C for 3–4 h. Cultures were spun down for 30 min at $3500 \times g$ and the pellets were frozen overnight at −20 °C. The next day, pellets were thawed and resuspended in PBS. The resuspended pellet was incubated at 4 °C head-over-head at 20 RPM for 2 h. Cultures were spun down for 20 min at $3500 \times g$ at 4 °C and the nanobodies were purified from the supernatant using a 1 mL HisTrap HP column (GE Healthcare, Chicago, Illinois, USA). The purity of the nanobodies was verified by sodiumdodecyl sulfate-polyacrylamide gel electrophoresis (SDS-PAGE) under reducing conditions (Bio-Rad, Hercules, California, USA).

**ELISA binding assay.** Fifty micrograms of membrane extracts of HEK293T-iUS28 or HEK293T cells transfected with the different US28 ICL mutants were coated overnight in a 96 well MicroWell MaxiSorp flat bottom plate (Sigma-Aldrich, Saint Louis, Missouri, USA). The next day, plates were washed with 1× PBS and blocked with 2% (w/v) skimmed milk in PBS for 1 h at RT. Nanobodies were diluted in 2% (w/v) skimmed milk and incubated for 1 h at RT. Nanobodies were detected with mouse-anti-Myc antibody (1:1000, Clone 9B11, Cell Signaling Technology, Leiden, The Netherlands) and horseradish peroxidase (HRP)-conjugated goat-anti-mouse antibody (1:1000, Bio-Rad). Incubations with antibodies were done for 1 h at RT. Wells were washed three times with 1× PBS between all incubation steps. Binding was determined with 1-step Ultra TMB-ELISA substrate (Thermo Fisher Scientific) and the reaction was stopped with 1 M H2SO4. Optical density was measured at 450 nm with a PowerWave plate reader (BioTek). Data were analyzed using GraphPad Prism version 8.0 (GraphPad Software, Inc., La Jolla, CA, USA).

**Immunofluorescence microscopy.** U251-iUS28 cells were seeded in poly-L-lysine (Sigma-Aldrich) coated 96-well plates and US28 expression was induced for 48 h at 37 °C and 5% CO2. In the case of transfected HEK293T cells, cells were seeded in poly-L-lysine (Sigma-Aldrich) coated 96-well plates 24 h post transfection and cells were incubated for 24 h before staining. To stain the extracellular binding, the medium of the cells was removed and the cells were incubated with 100 nM of the bivalent nanobodies for 1 h on ice. Cells were washed with 1× PBS and fixed with 4% paraformaldehyde (Sigma-Aldrich) for 10 min at room temperature (RT). Next, cells were washed and blocked with 1% PBS/FBS for 30 min at RT. After blocking, nanobodies were detected using Mouse-anti-Myc antibody (1:1000, 9B11 clone, Cell Signaling). Subsequently, cells were washed Goat-anti-Mouse Alexa Fluor 488 (1:1000 in 1% (v/v) FBS/PBS, Thermo Fisher Scientific). When binding of the nanobodies was tested on permeabilized cells, cells were first fixed and then permeabilized with 0.5% NP-40 in PBS (Sigma-Aldrich) for 30 min at RT. Next, cells were blocked and nanobodies were incubated for 1 h at RT. After incubation with nanobodies, the same incubation steps were performed using the primary and secondary antibodies. US28 expression was visualized with the rabbit-anti-US28 antibody (1:1000, Covance[40]) and with Goat-anti-Rabbit Alexa Fluor 488 or Goat-anti-Rabbit Alexa Fluor 546 (1:1000 in 1% (v/v) FBS /PBS, Thermo Fisher Scientific). Nuclei were stained using DAPI (Thermo Fisher Scientific). Cells were visualized with an Olympus FSX-100 microscope and 20x or 40x objective at RT. Data were acquired using Cell^B (Olympus) and FIJI/ImageJ version 1.51p was used for image analysis.

**Competition binding.** Increasing concentrations of nanobodies or unlabeled CX3CL1 or CCL5 were added to 100 pM $^{125}$I-CX3CL1 or 200 pM $^{125}$I-CCL5 in HEPES binding buffer (50 mM HEPES-HCl, pH 7.4; 1 mM CaCl2; 5 mM MgCl2; 0.1 M NaCl; 0.5% (w/v) bovine serum albumin). Radioligand alone and radioligand + 100 nM of unlabeled CX3CL1 or CCL5 were used as controls for total binding and non-specific binding. A total of 3 µg of HEK293T, HEK293T-iUS28 or

HEK293T cells transfected with the US28 receptor were added to the ligands and incubated for 2 h at room temperature. Membranes were harvested on 0.5%(w/v) PEI-soaked GF/C filter plates (Perkin-Elmer, Waltham, MA, USA) and dried for 30 min at 60 °C. In addition, 100 pM of $^{125}$I-CX3CL1 or 200 pM $^{125}$I-CCL5 was spotted on the GF/C filter plate to determine radioligand concentration. Scintillation fluid MicroScint-O (Perkin-Elmer) was added to the GF/C filter plate and radioactive decay was measured using a Microbeta liquid scintillation counter (Perkin-Elmer). Data were analyzed using GraphPad Prism version 8.0.

**Phospholipase C activation assay**. HEK293T cells were transfected with 100 ng of pcDEF3-HA-US28 and increasing amounts of pcDEF3-VUN103(-mVenus) or pcDEF3-non US28 targeting nanobody(-mVenus). In case of experiments with the addition of fractalkine, 1 µg of DNA encoding for the intrabody-mVenus constructs were co-transfected with 100 ng of pcDEF3-HA-US28. To assess the specificity of the intrabodies, HEK293T cells were transfected with 1 µg of DNA encoding the intrabody-mVenus constructs and 100 ng of pcDEF3-HA-Histamine 1 receptor. For the experiments with co-transfection of two intrabodies, cells were seeded in a 48-wells plate and transfected the next day with 50 ng encoding the intrabody-mVenus and 50 ng encoding the TEB-inducible intrabody-FLAG. After 6 h, tebufenozide (100 nM) was added to induce FLAG-tagged intrabody expression. To evaluate the effects of the intrabodies on US28 WT and US28 ICL2 CCR5 in U251 cells, U251-iUS28 cells were seeded in a 48-well plate and transfected the next day with 50 ng encoding the intrabody-mVenus. The next day, receptor expression was induced. For all setups, one day post-transfection cells were labeled in inositol-free DMEM (MP Biomedicals, Santa Ana, California, USA) supplemented with 1 µCi/ml myo-[2-$^3$H]inositol (Perkin Elmer) and incubated overnight at 37 °C and 5% CO$_2$. The next day; cells were subsequently incubated for with 10 mM LiCl in PBS, supplemented with 30 nM CX3CL1 for fractalkine experiments or 100 nM histamine for Histamine 1 receptor experiments. After 2 h, media was removed and cells were lysed with 10 mM cold formic acid. Inositol phosphates were isolated by anion-exchange chromatography (Dowex AG1-X8 columns, Bio-Rad) and quantified using a Packard Tri-Carb liquid scintillation counter (Perkin Elmer). Data were analyzed using GraphPad Prism version 8.0.

**Reporter gene assays**. HEK293T cells were transfected with 50 ng pcDEF3-HA-US28 VHL/E, 250 ng of pcDEF3-VUN103-mVenus or pcDEF3-non US28 targeting nanobody-mVenus and 2.5 µg reporter gene vector (NFAT, NF-kB or STAT3, Stratagene) as described above. Six hours post-transfection, cells were trypsinized using Trypsin-EDTA 0.05% (Gibco, Thermo Fisher Scientific) and 50,000 cells were seeded per well in a Poly-L-lysine treated white bottom 96-well assay plate and incubated at 37 °C and 5% CO$_2$. After 24 h, medium was removed and 25 µL LAR (0.83 mM D-Luciferine, 0.83 mM ATP, 0.78 µM Na$_2$HPO$_4$, 18.7 mM MgCl$_2$, 38.9 mM Tris-HCl (pH 7.8), 2.6 µM DTT, 0.03% Triton X-100 and 0.39% Glycerol) was incubated for 30 min at 37 °C. Luminescence (1 s per well) was measured using a CLARIOstar plate reader (BMG Labtech, Ortenberg, Germany). Data were analyzed using GraphPad Prism version 8.0.

**Determination of intrabody-mVenus expression levels**. On the day of the read-out for the experiments with the intrabody-mVenus constructs (e.g., phospholipase C assay), the relative expression of the different intrabodies was monitored by measuring the mVenus fluorescence at 540-20 nm using a CLARIOstar plate reader with excitation at 497-15 nm.

**Receptor and nanobody expression ELISA**. Transiently transfected HEK293T cells were seeded in a poly-L-lysine coated 96-wells plate for 48 h at 37 °C and 5% CO$_2$. Cells were fixed with 4% paraformaldehyde and permeabilized with 0.5% NP-40 as described above. In the case of determination of surface receptor expression levels, cells were only fixed with 4% paraformaldehyde. Receptor expression was detected using a Rat anti-HA antibody (1:1000 in 1% (v/v) FBS/PBS, Clone 3F10, Roche) and HRP-conjugated Goat-anti-Rat antibody (1:1000 in 1% (v/v) FBS/PBS, Pierce). Detection of intrabody-FLAG expression was done using a Mouse-anti-FLAG antibody (1:1000 in 1% (v/v) FBS/PBS, Clone M2, Sigma-Aldrich) and HRP-conjugated Goat-anti-Mouse antibody (1:2000 in 1% (v/v) FBS/PBS, Pierce). In the case of U251-iUS28 cells, receptor expression was detected using an Rabbit-anti-US28 antibody (1:1000 in 1% (v/v) FBS/PBS, Covance[24]) and HRP-conjugated Goat-anti-Rabbit antibody (1:1000 in 1% (v/v) FBS/PBS, Bio-Rad). Incubation with antibodies was done for 1 h at RT. Wells were washed three times with 1× PBS in between all incubation steps. Antibody binding was detected using 1-Step ultra TMB-ELISA substrate (Thermo Fisher Scientific) and the reaction was stopped with 1 M H$_2$SO$_4$. Optical density was measured at 450 nm with a PowerWave plate reader (BioTek). Data were analyzed using GraphPad Prism version 8.0.

**Detection of biotinylated proteins**. HEK293T cells were transfected with only 250 ng pcDEF3-US28 Bio ID2 or 250 ng pcDEF3-US28 Bio ID2 and 2 µg pcDEF3-nanobody-mVenus. During the same day of transfection, medium was replaced with medium containing a final concentration of 50 µM biotin. The next day, cells were lysed in native lysis buffer (25 mM Tris HCL pH7.4, 150 mM NaCl, 1 mM EDTA, 1% NP-40, 5% Glycerol, 1 mM NaF, 1 mM NaVO$_3$, cOmplete protease

inhibitor cocktail) for 10 min on ice. Cell debris was removed by centrifugation at 13,000 × g. Cell lysate was dialyzed overnight using Snakeskin Dialysis Tubing 10 K MWCO (Thermo Fisher Scientific) to remove the excess of biotin. Protein concentration of lysates was determined by Pierce BCA protein assay kit (Thermo Fisher Scientific) and same protein quantities were separated on a 10% SDS-PAGE gel under reducing conditions and transferred to 0.45 µm PVDF blotting membrane (GE healthcare, Chicago, IL, USA). Biotinylated proteins were detected using neutravidin-HRP (1:2000 in 5% BSA/TBS-T, A2664, Thermo Fisher Scientific). Blots were developed using Western Lightning Plus-ECL (Perkin-Elmer, Waltham, MA, USA) and visualized with Chemidoc (Bio-Rad).

**Pulldown and detection of biotinylated proteins**. Cell lysates were produced and dialyzed as described above. Streptavidin magnetic beads (New England Biolabs, Ipswich, MA, USA) were washed 3× with PBS and incubated with dialyzed cell lysate in a ratio of 250 µg lysate per 20 µL beads overnight at 4 °C head-over-head at 20 RPM. The next day, beads were washed twice with ice cold lysis buffer (50 mM Tris (pH = 7,5), 150 mM NaCl, 1%(w/v%) SDS, 1%(w/v%) Triton X-100, 0.5% (w/v%) NP-40, 10 µg/ml DNAse), once with ice cold 1 M KCL, once with 100 nM Na$_2$CO$_3$, once with 2 M Urea in 10 mM Tris pH = 8,0 and twice with lysis buffer. After washing, beads were resuspended in 1× Laemmli sample buffer supplemented with 100 mM Dithiothreitol (DTT) and 2 mM biotin. Samples were heated at 95 °C for 10 min and samples were removed from beads. Equal protein quantities (for lysates) or equal volumes of pulldown samples were separated on a 10% SDS-PAGE gel under reducing conditions and transferred to 0.45 µm PVDF blotting membrane. Pyruvate carboxylase was detected with a Rabbit-anti-pyruvate carboxylase antibody (1:500 in 5% skim milk/TBS-T, PA5-60552, Thermo Fisher Scientific). Nanobodies were detected using a Rabbit-anti-GFP antibody (1:1000 in 5% skim Milk/TBS-T, #2956, Cell Signaling Technology). Gα$_q$ was detected using a Rabbit-anti-Gα$_q$ antibody (1:1000 in 5% skimMilk/TBS-T, sc-393, Santa Cruz Biotechnology, Dallas, TX, USA). ß-arrestin 1/2 was detected using a Rabbit-anti-ß-arrestin 1/2 antibody (1:1000 in 5% BSA/TBS-T, #4674, Cell Signaling Technology). Actin was detected using a Mouse-anti-actin antibody (1:2000 in 5% skim Milk/TBS-T, Clone AC-74, Sigma-Aldrich). Pan 14-3-3 was detected using an anti-14-3-3 antibody (1:2000 in 5% skim milk/TBS-T, ab14108, Abcam). Antibodies were detected using Goat anti-Rabbit IgG-HRP conjugate (1:10000 in 5% skim Milk/TBS-T, #1706515, Bio-Rad) or Goat anti-Mouse IgG-HRP conjugate (1:10,000 in 5% skim Milk/TBS-T, #1706516, Bio-Rad). Blots were developed using Western Lightning Plus-ECL (Perkin-Elmer, Waltham, MA, USA) and visualized with Chemidoc (Bio-Rad) or autoradiography film. To normalize different pulldown samples, protein levels were normalized to pyruvate carboxylase and subsequently normalized to samples of cells only expressing HA-US28 Bio ID2 receptor.

**LC-MS/MS**. Peptides were separated using an Ultimate 3000 Nano LC-MS/MS system (Dionex LC-Packings, Amsterdam, The Netherlands) equipped with a 40-cm × 75-µm ID fused silica column custom packed with 1.9-µm, 120-Å ReproSil Pur C18 aqua (Dr Maisch GMBH, Ammerbuch-Entringen, Germany). After injection, peptides were trapped at 6 µl/min on a 10-mm × 100-µm ID trap column packed with 5-µm, 120-Å ReproSil Pur C18 aqua at 2% buffer B (buffer A: 0.5% acetic acid (Fischer Scientific), buffer B: 80% ACN, 0.5% acetic acid) and separated at 300 nl/min in a 10–40% buffer B gradient in 90 min (130 min inject-to-inject). Eluting peptides were ionized at a potential of +2 kV into a Q Exactive HF mass spectrometer (Thermo Fisher, Bremen, Germany). Intact masses were measured at resolution 120.000 (at m/z 200) in the Orbitrap using an AGC target value of 3E6 charges. The top 15 peptide signals (charge-states 2+ and higher) were submitted to MS/MS in the HCD (higher-energy collision) cell using 1.6 amu isolation width and 25% normalized collision energy. MS/MS spectra were acquired at resolution 15.000 (at m/z 200) in the orbitrap using an AGC target value of 1E6 charges, a maxIT of 32 ms and an underfill ratio of 0.1%. Dynamic exclusion was applied with a repeat count of 1 and an exclusion time of 30 s.

**Protein identification**. MS/MS spectra were searched against the Swissprot 2019-02 human canonical and isoform FASTA file (42417 entries) using MaxQuant version 1.6.4.0[69] supplemented with US28 Bio ID2 and nanobody construct sequences. Enzyme specificity was set to trypsin and up to two missed cleavages were allowed. Cysteine carboxamidomethylation (Cys, +57.021464 Da) was treated as fixed modification and methionine oxidation (Met, +15.994915 Da) and N-terminal acetylation (N-terminal, +42.010565 Da) as variable modifications. Peptide precursor ions were searched with a maximum mass deviation of 4.5 ppm and fragment ions with a maximum mass deviation of 20 ppm. Peptide and protein identifications were filtered at an FDR of 1% using the decoy database strategy. The minimal peptide length was 7 amino acids. Proteins that could not be differentiated based on MS/MS spectra alone were grouped to protein groups (default MaxQuant settings). Searches were performed with the label-free quantification option selected. Proteins were quantified by spectral counting. Spectral counts were normalized on the sum of the counts per sample and differential protein analysis between groups was performed using the beta-binomial test[70].

**Mini-Gαq and ßarr2 BRET recruitment assay**. HEK293T cells were transiently transfected, using the cell suspension protocol as described above, with 30 ng of pcDEF3-US28-NLuc, 150 ng of pcDEF3-Venus-mini-Gαq or pcDEF3-ßarr2-mVenus and 500 ng of pcDEF3-intrabody-FLAG constructs. After transfection, cells were seeded in a poly-L-lysine coated white 96-well plate. Two days-post transfection, medium was aspirated and cells were incubated with a final concentration of 10 µM of furimazine (NanoGlo Luciferase Assay System, Promega, Madison, Wisconsin, USA) in Hank's Buffered Saline Solution (HBSS, Gibco, Thermo Fisher Scientific) supplemented with 0.05% (w/v) BSA for 10 min at RT before readout. In case of agonist stimulation, 100 nM CX3CL1 was added 30 min prior to readout. Bioluminescence was measured using the PHERAstar plate reader at 475/30 nm and 535/30 nm. BRET signals were normalized to the signal of non-induced HEK293T cells expressing US28-Nluc and Venus-mini-Gαq or ßarr2-mVenus but no intrabody.

**Fluorescently labeled chemokine binding assay**. Membranes from HEK293T cells that express NLuc-US28 WT and Nb fusion constructs were prepared as described above. Having determined the protein concentration using the Pierce BCA protein assay kit (Thermo Fisher Scientific), 100 ng of these membranes were added to white 96-wells plates. Subsequently, increasing concentrations (0.1 pM–10 nM) of CX3CL1-Alexa647 or CCL5-Alexa647 (Almac Group, Craigavon, United Kingdom) in HBSS supplemented with 0.05% BSA were added to the wells. After 1 h incubation at RT, 10 µM of furimazine in HBSS supplemented with 0.05% BSA was added and binding was measured using the PHER-Astar plate reader at 460-80 nm/620 nm-LP.

**Nanobody-mVenus BRET assay**. HEK293T cells were transiently transfected, using the cell suspension protocol as described above, with only 10 ng of pcDEF3-US28-Rluc or 10 ng of pcDEF3-US28-Rluc and different amounts of pcDEF3-nanobody-mVenus constructs to obtain 1:2, 1:4 or 1:8 ratio. In case of kinetic BRET assays, cells were transfected with a 1:4 US28-Rluc/nanobody-mVenus ratio. After transfection, cells were seeded in a poly-L-lysine (Sigma-Aldrich) coated white 96-well plates (Greiner). Two days-post transfection, medium was aspirated and cells were incubated with a final concentration of 5 µM of Coelenterazine-h (Promega, Madison, Wisconsin, USA) in Hank's Balanced Salt Solution (HBSS, Sigma Aldrich) supplemented with 0.05% (w/v) Bovine Serum Albumin (BSA, Melford Laboratories Ltd) for 10 min at RT before readout. In case of the kinetic BRET assays, 30 nM CX3CL1 (PeproTech, London, UK) or 100 nM CCL5 (PeproTech) were added right before the readout. Binding was measured using the PHERAstar plate reader (BMG Labtech, Ortenberg, Germany) at 475/30 nm and 535/30 nm.

**Confocal microscopy**. HEK293T cells were transfected with 200 ng pcDEF3-HA-US28 VHL/E and 200 ng of pcDEF3-VUN103-mVenus, pcDEF3-Nb7-mVenus or pcDEF3-non-US28 targeting nanobody-mVenus. The next day, cells were seeded on Ibitreat µ-slide 8 well (Ibidi, Gräfelfing, Germany).24 h post seeding, the surface US28 population was staining by incubation on ice with an Alexa Fluor555 tagged anti-HA antibody in 1% FBS/PBS (1:1000, #26183-A555, Thermo Fisher Scientific) for 1 h. Cells were washed 3 times with cold PBS. After washing of the cells, 30 nM CX3CL1 in 1% FBS/PBS was added and incubated for 20 min at 37 °C and 5% CO₂. Cells were fixed with 4% PFA in 1% FBS/PBS for 10 min at RT. Cells were washed with PBS and nuclei were stained using DAPI (Thermo Fisher Scientific). Confocal laser scanning microscopy was performed on a Nikon A1R + microscope (Nikon, Amsterdam, The Netherlands) equipped with a 60 × 1.4 oil-immersion objective. mVenus and Alexa Fluor555 were irradiated using 488 and 561 nm laser lines, respectively, and detected using GaAsP PMTs. DAPI was irradiated using the 405 nm laser line and detected using a regular PMT. The samples were scanned using a Galvano scanner (Nikon) at 2048 × 2048 pixels, corresponding to 52 nm pixel size at 37 °C. NIS-Elements (Nikon) was used for image acquisition and FIJI/ImageJ version 1.51p was used for image analysis. Co-localization of US28 and nanobody-mVenus was determined as described previously using the JACoP Plugin[71].

**3D spheroid growth**. Inducible U251 cells were induced (if necessary) and 3×10^4 cells were seeded in a 96-well hanging drop plate (3D Biomatrix, Ann Arbor, Mi, USA). Two days later, spheroids were transferred to 6-well plated coated with 0.75% agarose in culture media (DMEM + heat inactivated FBS). Three days later, spheroids were imaged with an Olympus FSX-100 microscope and ×4 objective at RT. Data were acquired using Cell^B (Olympus) and FIJI/ImageJ version 1.51p was used for quantification of spheroid area.

**Western blot of HCMV-infected cells**. Cells were seeded one day prior to infection. Next day, The medium was removed and cells were infected with HCMV WT or HCMV ΔUS28 virus in serum free medium with a multiplicity of infection (MOI) of 4. The next day, medium was replaced with serum free medium. Four days post-infection, cells were lysed in native lysis buffer (25 mM Tris HCL pH7.4, 150 mM NaCl, 1 mM EDTA, 1% NP-40, 5% Glycerol, 1 mM NaF, 1 mM NaVO₃, cOmplete protease inhibitor cocktail) for 10 min on ice. Cell debris was removed by centrifugation at 13,000 × g. Protein concentration of lysates was determined by Pierce BCA protein assay kit (Thermo Fisher Scientific) and same protein

quantities were separated on a 10% SDS-PAGE gel under reducing conditions and transferred to 0.45 µm PVDF blotting membrane (GE healthcare). Total STAT3 was detected with an anti-STAT3 antibody (1:1000 in 5% skim Milk/TBS-T, clone 79D7, Cell signaling Technology). Phospho-STAT3 was detected with an anti-phospho-STAT3 antibody (1:1000 in 5% BSA/TBS-T, clone D3A7, Cell signaling Technology). Immediate early was detected with an anti-CMV immediate early antigen-antibody (1:1000 in 5% skim Milk/TBS-T, MAB810R, Millipore, Burlington, MA, USA). Nanobodies were detected using an anti-GFP antibody (1:1000 in 5% skim Milk/TBS-T, #2956, Cell Signaling Technology). Actin was detected using anti-actin antibody (1:2000 in 5% skim milk/TBS-T, Clone AC-74, Sigma-Aldrich). Antibodies were detected using Goat anti-Rabbit IgG-HRP conjugate (1:10,000 in 5% skim milk/TBS-T, #1706515, Bio-Rad) or Goat anti-Mouse IgG-HRP conjugate (1:10,000 in 5% skim milk/TBS-T, #1706516, Bio-Rad). Blots were developed using Western Lightning Plus-ECL (Perkin-Elmer) and visualized with Chemidoc (Bio-Rad).

**VEGF secretion ELISA of HCMV-infected cells**. Cells were infected as described above. Four days post-infection, the medium was harvested. Secreted VEGF levels were detected using Human VEGF standard TMB ELISA development kit according to the providers' protocol (#900-T10, Peprotech). Briefly, Rabbit-anti-Human VEGF antibody (1:200, #500-P10, Peprotech) in dilution buffer (0.05% Tween-20, 0.1% BSA/PBS) was coated overnight at RT in a ELISA microplate. The next day, wells were washed with wash buffer (0.05% Tween20/PBS) and blocked with block buffer (1% BSA/PBS). A dilution series was made from a stock of Human VEGF standard solution (1 µg/ml) and samples were diluted 5x in dilution buffer. Samples and diluted VEGF stock was added to the wells and incubated for 2 h at RT. Wells were subsequently washed with wash buffer and were incubated with biotinylated Rabbit Anti-Human VEGF detection antibody (1:800 in dilution buffer, #500-P10BT, Peprotech) for 2 h at RT. Wells were washed again with wash buffer and incubated with streptavidin-HRP solution for 30 min at RT. Wells were washed with wash buffer and VEGF levels were detected using TMB solution. The reaction was stopped using 1 M HCL and optical density was measured at 450 nm with a PowerWave plate reader (BioTek). Data were analyzed using GraphPad Prism version 8.0.

**Biological material availability**. The custom made anti-US28 antibody and our own US28 nanobodies (VUN100 and VUN103) are available to others upon request. The nanobody Nb7 has been described previously and data are available online[37].

**Reporting summary**. Further information on research design is available in the Nature Research Reporting Summary linked to this article.

## Data availability
Data supporting the findings of this manuscript are available from the corresponding author upon reasonable request. Source data are provided with this paper.

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

## Acknowledgements

We thank Jelle van den Bor for helping with the establishment of the TEB-inducible intrabody constructs. This work was supported by the Netherlands Organization for Scientific Research (NWO: Vici grant 016.140.657).

## Author contributions

T.D.G., R.H., M.H.S., and M.J.S. designed the study; T.D.G. generated nanobodies and characterized them, including displacement assays and spheroid experiments; T.D.G. and N.B. performed the Bio ID2 and BRET experiments; N.B. performed the virus experiments; T.D.G., N.B., and T.G. performed the signaling assays; T.D.G. and M.B. performed the confocal microscopy experiments; T.D.G., R.G.d.H., and S.P. performed the mass spectrometry experiments; S.P. analyzed the mass spectrometry results; T.D.G., N.B., R.H., K.C.G., H.L.P., C.R.J., M.H.S., and M.J.S. wrote and/or reviewed the manuscript.

## Competing interests

R.H. is affiliated with QVQ Holding BV, a CRO offering VHH services and molecules. All other authors declare no conflict of interest.
