## [Peer Review File · Nature Communications]

Reviewers' Comments:

Reviewer #1:

Remarks to the Author:

In this paper, the authors identified a selective nanobody (VUN103) that can bind to the intracellular region of a viral GPCR, US28. Via biochemical and pharmacological assays, the authors showed that VUN103 binds to a ligand-independent state, which is distinct from the active state, and could inhibit the Gq-mediated signaling. In the end, they showed that VUN103 could further inhibit the growth of glioblastoma spheroid and thus may have therapeutic potential in cancer therapy.

While nanobodies are emerging as very useful tools for academic research and clinical interventions, it is still not entirely clear how the nanobody identified here regulates the receptor signaling.

1. A major point in the paper is that VUN103 binds to a constitutively active, apo-conformation of US28. Another question remains to be answered here is whether VUN103 is an allosteric modulator of US28. If VUN103 stabilizes an active-like state, an increase of high-affinity binding sites for agonists should be expected. The K_d and B_{max} with or without VUN103 should be compared.

On the other hand, apo conformation is a mixture of inactive and active states. Based on the data, it does not exclude the possibility that VUN103 may bind to an inactive state receptor or non-selectively bind to the intracellular part of US28 irrelevant of inactive or active conformations. Either of the situations will have the same effects on Gq-mediated signaling. The author used competition binding assays (Fig. 4) to support that VUN103 does not recognize the inactive conformation. 1) They used US28-expressing membranes co-expressed with or without nanobodies. While this is hard to quantify how much VUN103 or Nb7 is actually bound to the receptor, a better way to do it is to add purified nanobodies directly to the membrane to see if there is an allosteric effect. Similar assays can be found from Huang et al., Nature 2015 and Staus et al., Nature 2016. 2) again the equilibrium value K_d for the radioligand 3H-CCL5 with or without nanobodies should also be provided given CCL5 is also an agonist for receptor US28.

2. In Fig. 2, the author first showed that VUN103 could inhibit Gq-mediated IP3 accumulation, then they detected decreased activation of NFAT, NF κ B, and STAT3. Does it mean here that activation of those factors is mediated by Gq signaling? How VUN103 affects arrestin coupling given that they may also have steric clash as seen in Gq. Since the authors already have the BRET system, an experiment that can be done is to measure interactions between US28 and Gq or arrestin in the presence of VUN103 to see if the latter differentially affects the G protein or arrestin coupling. Next, the author identified that ICL2 is essential for VUN103 binding to US28. This is important information, but unfortunately incomplete. Is ICL2 the only region responsible for interaction considering that nanobodies usually bind to GPCRs with a 3-D interface? If only ICL2 is critical, which residues are the key for US28-VUN103 interaction?

3. In Fig. 3C, it shows that lots of proteins have been biotinylated and captured, it is unclear that the authors only focused on Gq level, especially it has been shown in Fig.2. Is Gq the only one that shows the difference with or without VUN103? US28 has also coupled to G11, G12/13, and beta-arrestins, it will be helpful if the authors could provide more information.

4. In Fig. 5 and 2E, the authors compared the IP accumulation of VUN103 and Nb7. Nb7 does not affect IP accumulation in the absence of CX3CL1. This appears to contradict one of the conclusions from Miles et al., Elife 2018 that "Nb7 selectively binds to a subset of pre-existing, stable active-like conformations" and a complex structure of ligand-free US28/Nb7 has been obtained, which indicates that Nb7 can bind to US28 even without CX3CL1 and inhibit Gq coupling.

In line 246, it says CX3CL1 allosterically reduces VUN103 binding. A key question here would be if the agonist CX3CL1 induces receptor conformational changes that are not favorable for VUN103 binding or activation of the receptor by CX3CL1 recruits G proteins which compete with VUN103 binding.

5. In Fig.6, if reducing the basal activity or inhibiting the G protein signaling could inhibit tumor growth, wouldn't an inverse agonist be a better choice considering the delivery issue of nanobodies? Given that Nb7 also binds to apo receptor, it will be useful to compare VUN103 with Nb7, as well as existed inverse agonists, in the same experiments in Fig 6 to show the advantage of VUN103.

6. The Methods section should be significantly expanded to provide essential information about how experiments were conducted.

Line 324 should be mu-opioid, not delta.

Reviewer #2:

Remarks to the Author:

The authors describe the discovery of a Nanobody (VUN103) that binds the intracellular side of US28, displacing the Gaq protein, thus preventing US28 signaling and reducing the growth of tumor spheroids. Moreover, the authors claim that VUN103 is conformation selective and stabilizes the 'apo-conformation' of the receptor. Whereas the data that support the first conclusion are sound, there is only one experiment (figure 5E) that supports the second conclusion. Indeed, it has been shown that the dynamic range of GPCR activation can be significant and that allosteric Nanobodies, such as claimed in this manuscript, can have significant and reciprocal effects on the affinities of agonists and inverse agonists [1]. Moreover, all nanobodies described so far that convincingly stabilize a functional conformation of a GPCR showed significant allosteric effects on the binding affinities of orthosteric agonists/inverse agonists.

The authors seem to be aware of this weakness and are indeed very careful in their conclusive paragraph: 'In contrast, binding of CX3CL1 and other US28 ligands results in increased binding of Nb7 and a small but significant decrease of VUN103 binding to US28'. It surprises me however that the authors do not use the argument that US28 is indeed a receptor with a high constitutive activity to explain this observation? One would expect that the dynamic range for such receptors is lower, possibly explaining the small allosteric affect of VUN103 on agonist binding? The alternative explanation would be that VUN103 is not conformational selective but acts by competitive inhibition of G-protein binding. Finally, the authors describe orthosteric compounds and Nanobodies that act as inverse agonists of US28? It would be interesting to see if VUN103 has an allosteric effect on the affinity of these compounds? I recommend the authors to reformulate their conclusions accordingly.

Textual comment

Line 79-80: It is not clear from this sentence if small molecules and bivalent nanobodies act as inverse agonists for ligand-independent or ligand-dependent activities or for both activities? This in turn makes the hypothesis formulated between lines 80 and 82 speculative. If constitutive activation and ligand induced activation result from different conformations, one would expect different orthosteric compounds (agonists, inverse agonists) to differentially affect these different modes of signalling?

Staus et al (2016) Nature 535, 448-452

Jan Steyaert

Reviewer #3:

Remarks to the Author:

Reviewer comments: This manuscript reports on discovery and characterization of VUN103 as

conformation-specific nanobody to selectively blunt signaling of the viral chemokine receptor US28. It is very interesting, well written, and easy to read, although I sometimes struggled with the figures that were more complicated than implied by the manuscript text, that, after reading, does not leave you with questions but the figures do. Also, some claims are not entirely data-supported. Therefore I recommend to further strengthen this interesting paper by some additional experimentation and textual changes to enhance clarity and to make sure conclusions are entirely supported by data. Below please find many specific suggestions how to achieve this.

1. The title is very general and I would make it more specific by naming the receptor.
2. Introduction, lines 91,92: „While Nb7 stabilizes the ligand-bound, active conformation of US28, VUN103 appears to recognize a different, constitutively active, apo-conformation of US28.“ Please state clearly here whether stabilization of these distinct active conformations is followed by receptor inhibition. This does not come across completely clearly to me for Nb7. Inhibition of signaling is stated clearly in the abstract but I am ‚lost‘ in the Intro concerning this important aspect.
3. Introduction, lines 61,62 : „Some GPCRs, such as the human calcitonin receptor, human cannabinoid CB1 receptor, human β 2-adrenergic receptor (β 2-AR) and in particular the viral GPCR US28, also signal in a ligand independent manner“. I am not really sure that the beta2AR is a good example for ligand independent signaling. I think that particularly adrenergic receptors are very much ligand-controlled and don't fit well alongside with the viral GPCRs. I would use better examples for ligand-independent activity, truly constitutively active GPCRs (MC receptors, GPR3,6,12, Mas oncogene, ghrelin receptor etc). All these are better examples than beta2 which one could consider removing here.
4. Results, line 108: „One lead hit (VUN103) of this large family was selected,“ Please explain why this one was chosen. It is not obvious from Fig1A. Which were the criteria that led to this specific one? Please explain better for clarity.
5. Figure S1: please check if this has been mislabelled: top right panel should read HA and not US28, I think. I also find the legend extremely brief and needed to check the experimental section to see what has been done. It would be nice to add a bit more detail to also catch the method quicker.
6. Results, lines 114-115: „The observation that VUN103 could bind to US28 but did not displace US28 ligands suggested that VUN103 recognized an intracellular epitope on US28.“ This is one possibility, please also consider another one, that it binds but inhibits radioligand association and dissociation equally and thereby does not alter equilibrium binding. Such an action may occur from the extra- and intracellular side. For this reason, it is all the more important to explicitly state how microscopy has been done: with permeabilized or non-permeabilized cells in FigS1. The authors solve this problem quickly in Fig1E but as I see FigS1 before Fig1E, I am in doubt for a moment. I leave it up to the authors to make this a bit more clear because the intact cell/permeabilized cell experiment is the way to discriminate an extracellular allosteric action from an intracellular one.
7. I don't really understand Figure 1E: why do we see Nb VUN100 staining on the membrane and US28 staining inside the cell? Shouldn't they stain the same protein at the same location? Please explain. Why is US28 staining so different in the upper and lower image of Fig1E in live intact cells? How specific is the US28 antibody? Does it also stain non US28-expressing U251 cells? Why is this control not included? Why do Nanobody stainings differ in permeabilized cells? Both stain the receptor, why does receptor location look so distinct? Are there other images that allow to grasp the message easier? The text in the paper reads well and simple but I really struggle with the details in this figure. They cannot stay as they are without additional explanation. If there were better resolution images, please replace.
8. Figure 2A: the fact that an irrelevant antibody does not inhibit US28-mediated IP accumulation is a fine control but no real proof that VUN103 suppresses IP by specific action through US28. A more

convincing control would be to show that, in the same assay, VUN103 does not inhibit IP production of an unrelated Gq-coupled receptor.

9. Figure 2E: it would be nice and convincing if the authors showed that there is no difference in intracellular expression of Nb7mV and VUN103mV to rule out differences in intracellular nanobody abundance. Sure the authors have done such a control at some point to include it in the suppl section.

10. Figure S2: I would recommend to show averaged data, much nicer and convincing than representative experiments. If this cannot be done, please explain why.

11. Results, lines 152,153 „While VUN103 inhibited the constitutive activity of US28 Δ 300, it did not affect the constitutive activity of the US28 ICL2 chimera, indicating that ICL2 loop of US28 but not the cytoplasmic tail is important for binding of VUN103.“ To corroborate this conclusion it would be nice to not only replace ICL2 and use a tail-deleted construct but additionally use an ICL3 chimera and a tail swap chimera to figure out whether binding is exclusively mediated via ICL2. At present we can only conclude that ICL2 is involved but we don't know whether additional regions contribute as well. This is to learn whether intrabody action is including ICL2 or whether ICL2 is the only determinant. If this is 'too much', please make the limitations of the currently used chimera more clear.

12. Results section related to Figure 3: please state briefly why the US28-Bio ID2 construct displays lower basal activity. Is it functionally compromised due to the C-terminal fusion or is it just expressed at lower levels? Particularly if it was functionally compromised, this would affect the conclusions derived with the nanobodies. Therefore this distinction is not only helpful but also necessary to entirely support the author's claims.

13. Figure 3C: please look careful at this figure, either mock-transfected cells are missing (which I don't think) or the figure is mislabeled: + instead of minus in the US28 Bio ID2 lane. Please correct.

14. Figure 3D: this looks very challenging and I would like to see the other two westernblots. I very much like the data but need to see all experiments to better judge on the suitability of this method, which is used to derive a highly important conclusion: competition of nb with the G protein.

15. Figure 5A: Does CX3CL1 internalize US28 in the presence of the three nanobodies? Can this be quantified? I cannot draw a conclusion from the images and the text does not state anything specific in regard to this aspect. How does membrane abundance of US28 change in the presence of the three nanobodies and can this be quantified? This is important as I would expect less CX3CL1-induced internalization in the presence of VUN103. Alternatively, can the plasma membrane abundance be quantified as I would expect more US28 in the membrane when cells are exposed to CX3CL1 in the presence of VUN103. VUN103 should prevent internalization as compared with the irrelevant nanobody, presence of which should not impair the receptor from adopting the active conformation. Can the authors provide meaningful quantification of these images which only show a single/very few cells, so the 'bigger picture' is not clear. The authors focus only on the extent of colocalization of receptor and nanobody in the presence of ligand but don't comment in the main text on what the agonist does to the receptor under the three conditions, nor how membrane abundance of US28 changes. This is no less meaningful but the reader is left alone with the search for the answer and has to judge from one or two cells.

16. Figure 5E: this panel is extremely important but incomplete in its current form. It must be expanded by combining both nanobodies to achieve complete inhibition of CX3CL1-induced IP1 accumulation. These data will not only make a strong but also very relevant addition. I think this is a must-experiment to be provided with the revision. Can the authors please comment on the fact that ligand-induced IP accumulation is lower than basal IP production? How is this possible? I would expect this to be the other way round. Please explain. Statistics is missing in Fig5E. Is the Nb7 mV effect significant in the absence of CX3CL1?

17. Figure 6A: here it seems as if receptor and receptor-binding nanobody are at the same place, distinct from my concerns for Figure 1. Therefore again, could these 'not so ideal' stainings in Figure 1 be replaced by images that agree with Figure 6?

18. Results, lines 276-280: "To confirm that this effect was VUN103-mediated, US28 expression was measured. We saw no differences in US28 levels for the different transduced cell lines (Figure S5). Therefore, VUN103, upon lentiviral gene delivery, also blocks US28 signaling in glioblastoma cells and can inhibit US28-mediated glioblastoma spheroid growth." I appreciate the VUN103 effect but not the conclusion of the authors. This experiment does not confirm that the effect is VUN-mediated, it ONLY confirms that VUN does not act by lowering US28 expression, likely suggesting that it acts by lowering its function. However, please note that you have no causal proof for such a statement either, but correlative evidence only (VUN may also act via a US28-independent mechanism), so rephrase carefully to make sure that conclusions are entirely supported by data. An experiment unambiguously proving that VUN acts by specific inhibition of US28 in this spheroid growth model would be to show that it does not reduce spheroid size induced by a US28-CCR5-ICL2 chimera as used in Figure 2. If such a construct was expressed inducibly, it should cause VUN-resistant spheroid growth. I don't request this type of data but would like to make sure that precisely this type of experimentation would be needed to show a causal relation between VUN action and spheroid growth inhibition which would significantly strengthen the paper and its translational conclusion. However, based on the data provided the conclusion, "Therefore, VUN103, upon lentiviral gene delivery, also blocks US28 signaling in glioblastoma cells and can inhibit US28-mediated glioblastoma spheroid growth." is, unfortunately, not supported.

19. I also appreciate the data in Fig6D,E using the infection paradigm that VUN suppresses signaling outcomes when viruses are used that harbor US28. I also appreciate that VUN but not the irrelevant nanobody suppresses the signaling outcomes (pSTAT etc). Again, bringing a causal proof would significantly strengthen the data and the conclusion for example by using viruses with US28-CCR5-ICL2 which would be expected to induce pSTAT, secrete VEGF while being resistant to the action of VUN103. This type of experiment would really allow the conclusion that the newly identified intrabody does blunt oncosignaling by specific inhibition of US28. Without this kind of experimentation, the causal conclusion currently drawn is not supported because evidence at present is only correlative. The fact that the irrelevant nanobody has no effect does not prove that the action of VUN103 is specifically mediated via US28. In my view, these authors have the chance to significantly strengthen their manuscript by providing such suggested mechanistic evidence using their elegant construct with the CCR5-ICL2-swap.

20. Discussion, lines 325-326: "Nanobodies that stabilize an active GPCR conformation interact with the G protein interface and act as G protein mimetics." Would it be helpful to add, for clarity, that this action confers inhibition of signaling? So, like Nb7, which stabilizes the active conformation but kills G protein signaling.

21. Discussion 326-330: "Hence, the ICL2 is crucial for the recognition or stabilization of a GPCR-conformation in complex with a nanobody." This sentence sounds as if ICL2 is generally important for nb stabilization of GPCR conformations, however the specific example used here is US28. For the sake of balance and clarity, please make sure to specify that this is the case for US28, or provide additional examples for other GPCRs with related findings. Please also include the notion that ICL3 is no less important for signaling as compared with ICL2 for other GPCRs. Should there be a difference in contribution of ICL2 versus ICL3 concerning GPCR subfamilies, please briefly include in the discussion for clarity.

22. Discussion "However, our competition binding assays show no significant decrease in either affinity or total binding, making the second explanation less likely". I think your competition assay point into the right direction, even if data (affinity-decrease for agonists in the presence of VUN103) are not

significant. This also fits well to the small changes in the BRET assay in Fig5: the presence of agonist causes separation of US28 and the nanobody. In turn, occupancy of the receptor by the nanobody should stabilize a conformation that the agonist radioligand does not really like. I think this is what you see as a trend in your binding assays, but it is difficult to state without statistical significance.

23. Discussion, lines 337,338: "The effect of ligand binding on VUN103 binding and inhibition is small but significant." Maybe better to rephrase to 'the effect of VUN103 on ligand binding' OR 'the effect of VUN on ligand competition for the agonist-occupied receptor' or the like...'

24. Discussion, lines 360,361: "Full inhibition of US28 signaling could provide new therapeutic strategies to treat HCMV-positive tumors or latently-infected cells." This is a strong claim and I totally agree that data are tempting to make such a claim. However before we speculate about therapeutic strategies based on full inhibition of US28 signaling, it would be important to bring the causal proof that in the spheroid or infection paradigm VUN103 effects are indeed mediated by specific inhibition of US28. I made specific suggestions how to achieve this (see above).

Response to Reviewer letter

Manuscript: NCOMMS-20-03538-T

We thank the reviewers for their thorough and critical evaluation of our manuscript. We appreciate their comments and suggestions, which have helped improve our manuscript. We have addressed the comments point-by-point. We have enclosed a revised manuscript, marked up with the changes we have made. Please find the reviewers' comments with our answers in blue.

Reviewers' comments:

Reviewer #1 (Remarks to the Author):

In this paper, the authors identified a selective nanobody (VUN103) that can bind to the intracellular region of a viral GPCR, US28. Via biochemical and pharmacological assays, the authors showed that VUN103 binds to a ligand-independent state, which is distinct from the active state, and could inhibit the Gq-mediated signaling. In the end, they showed that VUN103 could further inhibit the growth of glioblastoma spheroid and thus may have therapeutic potential in cancer therapy.

While nanobodies are emerging as very useful tools for academic research and clinical interventions, it is still not entirely clear how the nanobody identified here regulates the receptor signaling.

A major point in the paper is that VUN103 binds to a constitutively active, apo-conformation of US28. Another question remains to be answered here is whether VUN103 is an allosteric modulator of US28. If VUN103 stabilizes an active-like state, an increase of high-affinity binding sites for agonists should be expected. The K_d and B_{max} with or without VUN103 should be compared. On the other hand, apo confirmation is a mixture of inactive and active states. Based on the data, it does not exclude the possibility that VUN103 may bind to an inactive state receptor or non-selectively bind to the intracellular part of US28 irrelevant of inactive or active conformations. Either of the situations will have the same effects on Gq-mediated signaling. The author used competition binding assays (Fig. 4) to support that VUN103 does not recognize the inactive conformation. 1) They used US28-expressing membranes co-expressed with or without nanobodies. While this is hard to quantify how much VUN103 or Nb7 is actually bound to the receptor, a better way to do it is to add purified nanobodies directly to the membrane to see if there is an allosteric effect. Similar assays can be found from Huang et al., Nature 2015 and Staus et al., Nature 2016. 2) again the equilibrium value K_d for the radioligand 3H-CCL5 with or without nanobodies should also be provided given CCL5 is also an agonist for receptor US28.

We thank the reviewer for his/her thorough evaluation of our manuscript and valuable feedback. We agree that additional experiments were required to substantiate our conclusion that VUN103 binds a constitutively active, apo-conformation of US28, while Nb7 favors binding to a ligand-bound active state. To support our conclusion we fused VUN103, Nb7 or a non-US28 targeting nanobody to the C-terminus of US28 and determined the binding properties of the chemokines CCL5 and CX3CL1. The fusion of nanobodies to the C-terminus of GPCRs provides a 1:1 ratio of GPCR and nanobody. This has previously been done to stabilize GPCRs in a certain conformation (Pardon et al., 2018, Angew Chem Int Ed Engl). In

order to quantify ligand binding directly and to do so non-radioactively, we have also inserted a Nanoluciferase (Nluc) at the N-terminus of US28 and determined the binding affinities of AlexaFluor647-labeled CX3CL1 and CCL5 using a BRET-based approach.

Our new results corroborate our previous results: binding of Nb7 to US28 increases the affinities of both CX3CL1 and CCL5 to US28, thus favoring binding to a ligand-bound active state of the receptor. This was not seen when US28 was fused to VUN103 or a non-US28 targeting nanobody (Figure 4C-D). Moreover, we observed an increase of surface-accessible US28 for ligand binding upon fusion of US28 with Nb7 and VUN103. This is in line with newly performed ELISA experiments and is explained by enhanced membrane localization of US28 when bound by either Nb7 or VUN103 (Figure 4E).

2. In Fig. 2, the author first showed that VUN103 could inhibit Gq-mediated IP3 accumulation, then they detected decreased activation of NFAT, NFkB, and STAT3. Does it mean here that activation of those factors is mediated by Gq signaling? How VUN103 affects arrestin coupling given that they may also have steric clash as seen in Gq. Since the authors already have the BRET system, an experiment that can be done is to measure interactions between US28 and Gq or arrestin in the presence of VUN103 to see if the latter differentially affects the G protein or arrestin coupling. Next, the author identified that ICL2 is essential for VUN103 binding to US28. This is important information, but unfortunately incomplete. Is ICL2 the only region responsible for interaction considering that nanobodies usually bind to GPCRs with a 3-D interface? If only ICL2 is critical, which residues are the key for US28-VUN103 interaction?

NFAT, NFkB and STAT3 signaling is indeed mediated by Gαq signaling. We have added the corresponding references.

To address the comments regarding the effect of VUN103 on Gαq and β-arrestin coupling, we have performed additional experiments (Figure 3F-I). We co-transfected HEK293T cells with US28-Nluc, mVenus-mini-Gαq or β-arrestin2-mVenus and inducible Flag-tagged VUN103 or non-US28 targeting nanobody. We then determined recruitment of mini-Gαq or β-arrestin2 to US28 by measuring BRET levels. Upon induction of intrabody expression, we observed a significant **decrease** in proximity between mVenus-mini-Gαq or β-arrestin2-mVenus and US28-Nluc. This was not the case for the irrelevant nanobody.

In addition, we have validated these results using the US28-Bio ID2 approach (Figure 3G-H). We detected increased levels of biotinylated β-arrestin1/2 upon US28-BioID2 expression and biotin addition. The results show constitutive recruitment of these proteins to US28. Moreover, US28-BioID2-mediated biotinylation of β-arrestin1/2 was decreased down to mock levels upon co-expression of VUN103-mVenus. The reduced interaction with β-arrestin proteins by VUN103 may contribute to the increases in surface expression of US28 upon co-expression of VUN103 (Figure 4E).

To further specify the binding epitope of VUN103, we have generated additional US28 mutants by swapping intracellular loops 1 and 3 of US28 with the corresponding loops of CCR5 (Figure S5). Moreover, we have included Nb7 as positive control, as the epitope (ICL2 and ICL3 of US28) recognized by this nanobody is already known from structural studies. While swapping ICL1 did not affect the affinity of

VUN103, when both ICL2 or ICL3 were exchanged for the corresponding ICL loops of CCR5 we saw no binding of VUN103. Both ICL2 and ICL3 are therefore important for VUN103 binding. Similar results were observed for Nb7.

3. In Fig. 3C, it shows that lots of proteins have been biotinylated and captured, it is unclear that the authors only focused on Gq level, especially it has been shown in Fig.2. Is Gq the only one that shows the difference with or without VUN103? US28 has also coupled to G11, G12/13, and beta-arrestins, it will be helpful if the authors could provide more information.

Our incentive for focusing on Gαq was to validate inhibition of US28-mediated Gαq signaling by VUN103. We provide evidence that Gαq recruitment to US28 is indeed blocked by VUN103 binding. However, Gαq was not the only biotinylated GPCR-interacting protein that was identified in the assay. We have included additional blots that show decreased biotinylation of β-arrestin1/2 upon co-expression of VUN103 (Figure 3G-H). As mentioned by the reviewer, there are many more GPCR-interacting proteins that might be influenced by VUN103 binding. An in depth analysis of the interactome of US28 is beyond the scope of this manuscript.

4. In Fig. 5 and 2E, the authors compared the IP accumulation of VUN103 and Nb7. Nb7 does not affect IP accumulation in the absence of CX3CL1. This appears to contradict one of the conclusions from Miles et al., Elife 2018 that “Nb7 selectively binds to a subset of pre-existing, stable active-like conformations” and a complex structure of ligand-free US28/Nb7 has been obtained, which indicates that Nb7 can bind to US28 even without CX3CL1 and inhibit Gq coupling.

We agree with the reviewer. In our view, our results do not contradict the paper of Miles et al. We point out that binding of Nb7 to apo US28 in the study of Miles et al. has been performed under different experimental conditions (non-cellular context with no competition from other proteins).

We have also determined the binding affinity of Nb7 to membrane extracts of HEK293T cells that overexpress apo US28. We saw that Nb7 can indeed bind to apo US28, albeit with low affinity (Figure S5). We have performed additional BRET experiments for US28-NLuc and mini-mVenus-Gαq construct upon co-expression of the different intrabodies (Figure 5D). Even in the absence of CX3CL1, both Nb7 and VUN103 decrease BRET signals between US28-NLuc and mVenus-mini-Gαq, compared to a non-US28 targeting nanobody. Competition between Gαq and the intrabodies suggest binding of Nb7 to apo US28. (Figure S9). Moreover, we show only a small but significant additional decrease of interaction between US28-NLuc and mVenus-mini-Gαq in the presence of CX3CL1 for Nb7 (Figure 5D). We have added these observations to the results section of the manuscript.

Based on these results, Nb7 might compete with Gαq for binding to apo-US28. If the affinity of Gαq for US28 is higher, then Nb7 would not be able to compete with Gαq for binding to apo-US28.

In line 246, it says CX3CL1 allosterically reduces VUN103 binding. A key question here would be if the agonist CX3CL1 induces receptor conformational changes that are not favorable for VUN103 binding or activation of the receptor by CX3CL1 recruits G proteins which compete with VUN103 binding.

We have performed additional BRET experiments between US28-NLuc and mVenus-mini-Gαq constructs upon co-transfection with the different intrabodies (Figure 3F and Figure 5D). Together with our previous experiment using US28-Rluc and mVenus-tagged intrabodies (Figure 5C), these results show that activation of the receptor by CX3CL1 recruits the Gαq protein, which competes for binding of VUN103.

5. In Fig.6, if reducing the basal activity or inhibiting the G protein signaling could inhibit tumor growth, wouldn't an inverse agonist be a better choice considering the delivery issue of nanobodies? Given that Nb7 also binds to apo receptor, it will be useful to compare VUN103 with Nb7, as well as existed inverse agonists, in the same experiments in Fig 6 to show the advantage of VUN103.

We agree with the reviewer. Finding full inverse agonists that target US28 extracellularly might potentially be more straightforward than the use of intrabodies. However, no other full inverse agonists with the desired properties (high affinity and specificity) have been developed to date. As gene delivery methods are evolving, we feel that there is a realistic possibility for therapeutic potential of VUN103 intrabody.

A comparison of the therapeutic effect Nb7 with VUN103 in a spheroid or viral setting would indeed be interesting. However, establishing stable cell lines with similar intrabody expression levels has proven to be challenging and time-consuming. Instead, we compared Nb7 and VUN103 in transiently transfected HEK293T cells (Figure 6A). As an additional experiment, we also evaluated the effect of co-expression of both intrabodies on US28-mediated signaling in the absence and presence of CX3CL1 (Figure 6A). Co-expression of both intrabodies could be an even more interesting approach to block US28, as this combination completely inhibited both the apo conformation and CX3CL1-bound conformation.

With regards to comparison the intrabodies with existing inverse agonists, we have previously performed comparable spheroid growth experiments with the partially inverse agonistic bivalent US28 nanobodies (Heukers et al., 2018, Oncogene), which resulted in 50% inhibition of US28-enhanced spheroid growth. We have added this reference to the manuscript.

6. The Methods section should be significantly expanded to provide essential information about how experiments were conducted.

We have expanded the Methods section as requested by the reviewer. However, we have tried to keep this concise, in order to adhere to the guideline of Nature Communications. We have cited previous publications in which the methods have been described more extensively.

Line 324 should be mu-opioid, not delta.

We have changed this mistake accordingly.

Reviewer #2 (Remarks to the Author):

The authors describe the discovery of a Nanobody (VUN103) that binds the intracellular side of US28, displacing the Gαq protein, thus preventing US28 signaling and reducing the growth of tumor spheroids. Moreover, the authors claim that VUN103 is conformation selective and stabilizes the 'apo-conformation' of the receptor. Whereas the data that support the first conclusion are sound, there is only one experiment (figure 5E) that supports the second conclusion. Indeed, it has been shown that the dynamic range of GPCR activation can be significant and that allosteric Nanobodies, such as claimed in this manuscript, can have significant and reciprocal effects on the affinities of agonists and inverse agonists [1]. Moreover, all nanobodies described so far that convincingly stabilize a functional conformation of a GPCR showed significant allosteric effects on the binding affinities of orthosteric agonists/inverse agonists.

The authors seem to be aware of this weakness and are indeed very careful in their conclusive paragraph: 'In contrast, binding of CX3CL1 and other US28 ligands results in increased binding of Nb7 and a small but significant decrease of VUN103 binding to US28'. It surprises me however that the authors do not use the argument that US28 is indeed a receptor with a high constitutive activity to explain this observation? One would expect that the dynamic range for such receptors is lower, possibly explaining the small allosteric effect of VUN103 on agonist binding? The alternative explanation would be that VUN103 is not conformational selective but acts by competitive inhibition of G-protein binding. Finally, the authors describe orthosteric compounds and Nanobodies that act as inverse agonists of US28? It would be interesting to see if VUN103 has an allosteric effect on the affinity of these compounds? I recommend the authors to reformulate their conclusions accordingly.

We appreciate this useful feedback and the suggestions for improvement. We agree with the reviewer that the dynamic range for GPCRs of the highly constitutive conformation, such as US28, is indeed relatively small. By performing additional experiments (Fig. 5C-D), we can now conclude that CX3CL1 binding indeed results in a significant increase of G protein binding and less VUN103 binding. Co-expression of Nb7 and VUN103 fully inhibits US28 activation, both in the absence and presence of CX3CL1. These nanobodies must therefore recognize different US28 conformations of US28.

Moreover, we determined the binding affinities of AlexaFluor647-labeled ligands CX3CL1 and CCL5 for NanoLuc-tagged US28-Intrabody fusion constructs by measuring BRET. These experiments corroborated our previous competition binding studies using radiolabeled ligands and show that VUN103 has no effect on the affinity of CX3CL1 and CCL5.

It would indeed be interesting to see the effect of Nb7 and VUN103 on inverse agonists. We have done experiments with VUF2274, the first described US28 inverse agonist. In our assay, using the Nb7/VUN103 fused receptor constructs, we observed that Nb7 binding increases the affinity of VUF2274 to US28, while no effect on binding was seen for VUN103 (Figure 1). These findings support the fact that some US28-targeting small molecule inverse agonists, including VUF2274, act as "camouflaged agonists" (Waldhoer et al., JBC, 2003; Tschammer N., Bioorg Med Chem Lett., 2014). We consider this particular point to be

beyond the scope of the paper. We have included this data in the response to reviewers and not in the paper.

Figure 1. Displacement of A647-CX3CL1 by VUF2274. BRET-based affinity determination of AlexaFluor (A)647-labeled CX3CL1 using membrane extracts of HEK293T cells expressing wildtype NLuc-US28 or NLuc-US28-nanobody fusions. In the case of the different NLuc-US28-nanobody fusions, a non-US28 targeting nanobody (Irr Nb), VUN103 or Nb7 was genetically fused to the C-terminus of NLuc-US28. Binding was determined by subtraction of the background (buffer only) and values were normalized to receptor expression. Percentage of binding was calculated by setting the CX3CL1 binding at highest concentration to US28 WT as 100% binding. pKi values were determined using the affinity of A647-CX3CL1 for the different constructs as determined in Figure 4 and Table 2 in the manuscript.

We did show that the extracellularly binding and antagonistic nanobodies act as inverse agonists in bivalent formats. This might be explained by uncoupling of G proteins by receptor dimerization, as suggested for other inverse (bivalent) nanobodies that target chemokine receptors. As this effect is due to dimerization and not conformational change, these molecules cannot be used to determine the effect of inverse agonists.

We have therefore reformulated our conclusions and added a comment to the discussion section that our results are most likely attributable to a smaller dynamic range of US28.

Textual comment

Line 79-80: It is not clear from this sentence if small molecules and bivalent nanobodies act as inverse agonists for ligand-independent or ligand-dependent activities or for both activities? This in turn makes the hypothesis formulated between lines 80 and 82 speculative. If constitutive activation and ligand

induced activation result from different conformations, one would expect different orthosteric compounds (agonists, inverse agonists) to differentially affect these different modes of signalling?

We apologize for any unclarity in our formulation of this statement. We have changed our sentence to: "Most receptor modulators bind to the extracellular portion of US28 and inhibit its activity by acting as antagonists and/or inverse agonists."

The reviewer is correct in his reasoning. Different effects on signaling are to be expected. The reported modulators can be subdivided into those that act as neutral antagonists, solely inhibiting ligand-dependent signaling or inverse agonists, which inhibit both ligand-dependent signaling and independent-signaling or inhibit only ligand-independent signaling. Moreover, some of the reported inverse agonistic molecules (VUF2274) act as non-competitive antagonists, further indicating potential different receptor conformations. See also our comments above. Although the binding and/or functional effects of these different compounds to different conformational states is still a field of investigation, these data indicate that there could be differences between different conformations.

Reviewer #3 (Remarks to the Author):

Reviewer comments: This manuscript reports on discovery and characterization of VUN103 as conformation-specific nanobody to selectively blunt signaling of the viral chemokine receptor US28. It is very interesting, well written, and easy to read, although I sometimes struggled with the figures that were more complicated than implied by the manuscript text, that, after reading, does not leave you with questions but the figures do. Also, some claims are not entirely data-supported. Therefore I recommend to further strengthen this interesting paper by some additional experimentation and textual changes to enhance clarity and to make sure conclusions are entirely supported by data. Below please find many specific suggestions how to achieve this.

We thank the reviewer for his/her feedback and suggestions for improvement. We have improved the figures to make them less complicated, as suggested by the reviewer. Moreover, we have added additional controls and experimental data to support our claims.

1. The title is very general and I would make it more specific by naming the receptor.

We have added "US28" to the title.

2. Introduction, lines 91,92: „While Nb7 stabilizes the ligand-bound, active conformation of US28, VUN103 appears to recognize a different, constitutively active, apo-conformation of US28.“ Please state clearly here whether stabilization of these distinct active conformations is followed by receptor inhibition. This does not come across completely clearly to me for Nb7. Inhibition of signaling is stated clearly in the abstract but I am ‚lost‘ in the Intro concerning this important aspect.

We have clarified this in the introduction. We have changed these sentences to: “ Nb7 stabilizes the ligand-bound, active conformation of US28 and inhibits ligand-induced signaling. VUN103 appears to recognize and inhibit a different, constitutively active, apo-conformation of US28.”

3. Introduction, lines 61,62 : „Some GPCRs, such as the human calcitonin receptor, human cannabinoid CB1 receptor, human β 2-adrenergic receptor (β 2-AR) and in particular the viral GPCR US28, also signal in a ligand independent manner“. I am not really sure that the beta2AR is a good example for ligand independent signaling. I think that particularly adrenergic receptors are very much ligand-controlled and don't fit well alongside with the viral GPCRs. I would use better examples for ligand-independent activity, truly constitutively active GPCRs (MC receptors, GPR3,6,12, Mas oncogene, ghrelin receptor etc). All these are better examples than beta2 which one could consider removing here.

We agree that the human β 2-adrenergic receptor is not the best example of constitutive activity. We have removed this example and now refer to the melanocortin and ghrelin receptors instead.

4. Results, line 108: „One lead hit (VUN103) of this large family was selected,“ Please explain why this one

was chosen. It is not obvious from Fig1A. Which were the criteria that led to this specific one? Please explain better for clarity.

We have performed preliminary screenings on different members of the family to find the clone with the highest affinity. We have added this information to the manuscript.

5. Figure S1: please check if this has been mislabelled: top right panel should read HA and not US28, I think. I also find the legend extremely brief and needed to check the experimental section to see what has been done. It would be nice to add a bit more detail to also catch the method quicker.

This is indeed a mistake. We have changed this accordingly. We have expanded the legend as well.

6. Results, lines 114-115: „The observation that VUN103 could bind to US28 but did not displace US28 ligands suggested that VUN103 recognized an intracellular epitope on US28.“ This is one possibility, please also consider another one, that it binds but inhibits radioligand association and dissociation equally and thereby does not alter equilibrium binding. Such an action may occur from the extra- and intracellular side. For this reason, it is all the more important to explicitly state how microscopy has been done: with permeabilized or non-permeabilized cells in FigS1. The authors solve this problem quickly in Fig1E but as I see FigS1 before Fig1E, I am in doubt for a moment. I leave it up to the authors to make this a bit more clear because the intact cell/permeabilized cell experiment is the way to discriminate an extracellular allosteric action from an intracellular one.

We acknowledge the possibility that VUN103 does not alter equilibrium binding of US28 ligands. However, we show intracellular binding of VUN103 by fluorescence microscopy, as mentioned by the reviewer. We apologize for not being completely clear. We have changed the order of the experiments shown in Figure 1 to avoid potential misunderstanding. Moreover, we have clarified the use of permeabilized/non-permeabilized cells in the text and legends.

7. I don't really understand Figure 1E: why do we see Nb VUN100 staining on the membrane and US28 staining inside the cell? Shouldn't they stain the same protein at the same location? Please explain. Why is US28 staining so different in the upper and lower image of Fig1E in live intact cells? How specific is the US28 antibody? Does it also stain non US28-expressing U251 cells? Why is this control not included? Why do Nanobody stainings differ in permeabilized cells? Both stain the receptor, why does receptor location look so distinct? Are there other images that allow to grasp the message easier? The text in the paper reads well and simple but I really struggle with the details in this figure. They cannot stay as they are without additional explanation. If there were better resolution images, please replace.

We apologize for the misunderstanding with regards to the microscopy data. We have addressed this by performing additional fluorescence microscopy experiments. These further clarify the results by separately showing US28 expression and nanobody binding.

To clarify our previous set-up, the polyclonal anti-US28 antibody binds specifically to intracellular epitopes of US28. For the binding of the nanobodies to the non-permeabilized cells, we incubated cells with the nanobodies on ice, preventing internalization and thus resulting in staining of surface US28 only. Next, we permeabilized the cells to enable intracellular staining of US28 with a polyclonal anti-US28 antibody. The intracellular portion of the US28 population is stained with the US28 antibody but not with the VUN100 nanobody. As 80% of the US28 population is located intracellularly (due to constitutive internalization of the receptor), VUN100 only binds a small fraction of the total receptor pool on the cell surface. In contrast, the vast majority of the staining signal comes from the intracellular receptor pool stained by the polyclonal anti-US28 antibodies.

To further clarify that VUN103 binds US28 intracellularly, we have performed additional immunofluorescent microscopy experiments in which we separately stained for US28 to show that US28 is expressed (Figure 1C) on the cells. We performed a subsequent staining with only the nanobodies (Figure 1D).

8. Figure 2A: the fact that an irrelevant antibody does not inhibit US28-mediated IP accumulation is a fine control but no real proof that VUN103 suppresses IP by specific action through US28. A more convincing control would be to show that, in the same assay, VUN103 does not inhibit IP production of an unrelated Gq-coupled receptor

We thank the reviewer for this suggestion. We have performed additional experiments showing that VUN103 does not have an effect on the histamine-induced H1 receptor mediated inositol phosphate accumulation.

We have addressed this in the results section and added a supplemental figure (Figure S2).

9. Figure 2E: it would be nice and convincing if the authors showed that there is no difference in intracellular expression of Nb7mV and VUN103mV to rule out differences in intracellular nanobody abundance. Sure the authors have done such a control at some point to include it in the suppl section.

We performed this control experiment and have now added it to the manuscript (Figure S3).

10. Figure S2: I would recommend to show averaged data, much nicer and convincing than representative experiments. If this cannot be done, please explain why.

While we agree with the reviewer, it is not possible to show averaged data (now Figure S4) due to the experimental procedures. In the course of ELISA experiments, the reaction is quenched by addition of H₂SO₄. This is performed manually; timing of the reaction is based on the color change observed in the control samples). Therefore the raw OD values differ between the different experiments. Therefore we cannot average the datapoints of the different assays.

11. Results, lines 152,153 „While VUN103 inhibited the constitutive activity of US28 Δ 300, it did not affect the constitutive activity of the US28 ICL2 chimera, indicating that ICL2 loop of US28 but not the cytoplasmic

tail is important for binding of VUN103.” To corroborate this conclusion it would be nice to not only replace ICL2 and use a tail-deleted construct but additionally use an ICL3 chimera and a tail swap chimera to figure out whether binding is exclusively mediated via ICL2. At present we can only conclude that ICL2 is involved but we don't know whether additional regions contribute as well. This is to learn whether intrabody action is including ICL2 or whether ICL2 is the only determinant. If this is 'too much', please make the limitations of the currently used chimera more clear.

As requested also by reviewer 1, we have generated additional US28 mutants where we have swapped the ICL1 and ICL3 of US28 for the corresponding ICL loops of CCR5 (Figure S5). Swapping of the ICL1 loop did not significantly affect the affinity of VUN103. In contrast, VUN103 did not bind US28 with ICL2 or ICL3 from CCR5, indicating that both ICL2 and ICL3 are involved in binding of VUN103. We have included Nb7 as a positive control, as the binding epitope (ICL2 and ICL3) of this nanobody to US28 was known from structural studies. Similar results, as seen for VUN103, were observed for Nb7.

12. Results section related to Figure 3: please state briefly why the US28-Bio ID2 construct displays lower basal activity. Is it functionally compromised due to the C-terminal fusion or is it just expressed at lower levels? Particularly if it was functionally compromised, this would affect the conclusions derived with the nanobodies. Therefore this distinction is not only helpful but also necessary to entirely support the author's claims.

This is a valid point. Similar expression levels were seen with the different constructs, indicating that tagging with the Bio ID2 does not affect expression but does influence the functionality of US28 at least in part. Nevertheless, we could still detect binding and inhibition by VUN103.

To validate our Bio ID2 results, we performed additional BRET experiments using different US28 constructs (Figure 3F and I), which confirm that VUN103 binding results in impaired G protein and β -arrestin coupling to US28.

13. Figure 3C: please look careful at this figure, either mock-transfected cells are missing (which I don't think) or the figure is mislabeled: + instead of minus in the US28 Bio ID2 lane. Please correct.

We had included the mock samples in our western blot experiment but then removed them from the figure for clarity. We have now reinstated them as referred to in the manuscript.

14. Figure 3D: this looks very challenging and I would like to see the other two westernblots. I very much like the data but need to see all experiments to better judge on the suitability of this method, which is used to derive a highly important conclusion: competition of nb with the G protein.

The reviewer is correct . Additional experiments are required to support our findings. To provide further evidence that VUN103 actually competes for G protein binding, we have performed additional BRET experiments between US28-Nluc and mVenus-mini-G α q constructs upon co-transfection with the

different intrabodies (Figure 3F). These experiments validate our immunoblot results and show a decrease of interaction between US28-Nluc and mVenus-mini-Gαq upon co-expression of VUN103.

15. Figure 5A: Does CX3CL1 internalize US28 in the presence of the three nanobodies? Can this be quantified? I cannot draw a conclusion from the images and the text does not state anything specific in regard to this aspect. How does membrane abundance of US28 change in the presence of the three nanobodies and can this be quantified? This is important as I would expect less CX3CL1-induced internalization in the presence of VUN103. Alternatively, can the plasma membrane abundance be quantified as I would expect more US28 in the membrane when cells are exposed to CX3CL1 in the presence of VUN103. VUN103 should prevent internalization as compared with the irrelevant nanobody, presence of which should not impair the receptor from adopting the active conformation. Can the authors provide meaningful quantification of these images which only show a single/very few cells, so the 'bigger picture' is not clear. The authors focus only on the extent of colocalization of receptor and nanobody in the presence of ligand but don't comment in the main text on what the agonist does to the receptor under the three conditions, nor how membrane abundance of US28 changes. This is no less meaningful but the reader is left alone with the search for the answer and has to judge from one or two cells.

This is a valid point. We have therefore performed additional ELISA experiments to address this (examining surface and total US28 expression levels upon nanobody binding (Figure 4E). These results show an increase of total and surface levels of US28 upon co-expression of VUN103 and Nb7. This we attribute to an inhibition of internalization and degradation of US28, a continuous process regulated by its constitutive activity.

Moreover, we have performed additional BRET experiments with AlexaFluor647-labeled CX3CL1 or CCL5 and NanoLuc-tagged US28-intrabody fusion constructs. Such fusion of nanobodies to the C-terminus of GPCRs has previously been done by Pardon et al. to stabilize GPCRs in a certain conformation. This approach enforces a 1:1 ratio of GPCR and nanobody. Both Nb7 and VUN103 fused to the C-terminus of US28 resulted in an increase of overall BRET levels, in line with increases in surface and total expression. It fits our additional ELISA experiments (Figure 4C-D).

Finally, we performed additional Bio-ID-western blot experiments to determine the effect of VUN103 on coupling of β -arrestin1/2 to US28. Expression of US28-Bio ID2 resulted in biotinylation of β -arrestin1/2 (Figure 3G-H). Co-expression of VUN103 results in lower levels of biotinylated β -arrestin1/2, indicating that VUN103 binding inhibits recruitment of β -arrestin1/2 to US28. This impaired coupling of β -arrestin1/2 might inhibit US28 internalization and degradation. It would explain the differences in (surface) expression and internalization observed in the other experiments.

16. Figure 5E: this panel is extremely important but incomplete in its current form. It must be expanded by combining both nanobodies to achieve complete inhibition of CX3CL1-induced IP1 accumulation. These data will not only make a strong but also very relevant addition. I think this is a must-experiment to be provided with the revision. Can the authors please comment on the fact that ligand-induced IP

accumulation is lower than basal IP production? How is this possible? I would expect this to be the other way round. Please explain. Statistics is missing in Fig5E. Is the Nb7 mV effect significant in the absence of CX3CL1?

We agree with the reviewer and have performed the suggested experiment (Figure 6). We show that co-expression of Nb7 and VUN103 results in complete inhibition of US28 signaling, both in the absence and presence of CX3CL1. We failed to see such complete inhibition when co-expressing an irrelevant nanobody with VUN103 or with Nb7.

With regards to the effect of CX3CL1 on inositol phosphate accumulation, we do not see a significant drop in inositol phosphate accumulation upon addition of CX3CL1 (Figure 2E and 6). We have added statistics to these figures to make this point more clearly as well. Our results are also in line with the paper of Miles et al. when using HEK293T cells (Miles et al., eLife, 2018).

17. Figure 6A: here it seems as if receptor and receptor-binding nanobody are at the same place, distinct from my concerns for Figure 1. Therefore again, could these 'not so ideal' stainings in Figure 1 be replaced by images that agree with Figure 6?

We have changed the stainings from Figure 1 as mentioned in our answer to the previous comments.

18. Results, lines 276-280: "To confirm that this effect was VUN103-mediated, US28 expression was measured. We saw no differences in US28 levels for the different transduced cell lines (Figure S5). Therefore, VUN103, upon lentiviral gene delivery, also blocks US28 signaling in glioblastoma cells and can inhibit US28-mediated glioblastoma spheroid growth." I appreciate the VUN103 effect but not the conclusion of the authors. This experiment does not confirm that the effect is VUN-mediated, it ONLY confirms that VUN does not act by lowering US28 expression, likely suggesting that it acts by lowering its function. However, please note that you have no causal proof for such a statement either, but correlative evidence only (VUN may also act via a US28-independent mechanism), so rephrase carefully to make sure that conclusions are entirely supported by data. An experiment unambiguously proving that VUN acts by specific inhibition of US28 in this spheroid growth model would be to show that it does not reduce spheroid size induced by a US28-CCR5-ICL2 chimera as used in Figure 2. If such a construct was expressed inducibly, it should cause VUN-resistant spheroid growth. I don't request this type of data but would like to make sure that precisely this type of experimentation would be needed to show a causal relation between VUN action and spheroid growth inhibition which would significantly strengthen the paper and its translational conclusion. However, based on the data provided the conclusion, "Therefore, VUN103, upon lentiviral gene delivery, also blocks US28 signaling in glioblastoma cells and can inhibit US28-mediated glioblastoma spheroid growth." is, unfortunately, not supported.

We agree with the reviewer. We therefore performed additional inositol phosphate accumulation experiments using U251 cell lines that stably express HA-tagged US28 wildtype or HA-tagged US28-CCR5-ICL2 receptor (Figure S11). These cell lines were transiently transfected with irrelevant nanobody-mVenus or VUN103-mVenus constructs and accumulation of inositol phosphate was determined. Whereas VUN103-mVenus did inhibit US28 wildtype, it was not able to inhibit accumulation of inositol phosphate

by US28-CCR5-ICL2. Unfortunately, we did not fully block US28 with VUN103 in this setting, which is due to low transfection efficiencies in these cells. However, it shows the lack of effect of VUN103 on US28 signaling in case of the ICL2 mutant.

19. I also appreciate the data in Fig6D,E using the infection paradigm that VUN suppresses signaling outcomes when viruses are used that harbor US28. I also appreciate that VUN but not the irrelevant nanobody suppresses the signaling outcomes (pSTAT etc). Again, bringing a causal proof would significantly strengthen the data and the conclusion for example by using viruses with US28-CCR5-ICL2 which would be expected to induce pSTAT, secrete VEGF while being resistant to the action of VUN103. This type of experiment would really allow the conclusion that the newly identified intrabody does blunt oncosignaling by specific inhibition of US28. Without this kind of experimentation, the causal conclusion currently drawn is not supported because evidence at present is only correlative. The fact that the irrelevant nanobody has no effect does not prove that the action of VUN103 is specifically mediated via US28. In my view, these authors have the chance to significantly strengthen their manuscript by providing such suggested mechanistic evidence using their elegant construct with the CCR5-ICL2-swap.

This is a valid point. However, the construction of a HCMV virus variant that contains the US28-CCR5-ICL2 mutant is challenging and time-consuming. Instead, we have now included data showing that VUN103 expression does not affect constitutive signaling of the US28-CCR5-ICL2 mutant, as inferred from inositol phosphate levels comparable to those seen for WT US28 (see Fig. S11A). Using this chimeric mutant we have now shown that the VUN103-mediated inhibition is US28-specific.

20. Discussion, lines 325-326: “Nanobodies that stabilize an active GPCR conformation interact with the G protein interface and act as G protein mimetics.” Would it be helpful to add, for clarity, that this action confers inhibition of signaling? So, like Nb7, which stabilizes the active conformation but kills G protein signaling.

We have changed the sentence to: “Nanobodies that stabilize an active GPCR conformation interact with the G protein interface and act as G protein mimetics, thereby inhibiting G protein mediated signaling”.

21. Discussion 326-330: “Hence, the ICL2 is crucial for the recognition or stabilization of a GPCR-conformation in complex with a nanobody.” This sentence sounds as if ICL2 is generally important for nb stabilization of GPCR conformations, however the specific example used here is US28. For the sake of balance and clarity, please make sure to specify that this is the case for US28, or provide additional examples for other GPCRs with related findings. Please also include the notion that ICL3 is no less important for signaling as compared with ICL2 for other GPCRs. Should there be a difference in contribution of ICL2 versus ICL3 concerning GPCR subfamilies, please briefly include in the discussion for clarity.

The reviewer is correct. The additional experiments on the newly generated US28 ICL1 and ICL3 mutants (Figure S5) indicate the importance of ICL3 in addition to ICL2 for nanobody stabilization. These new data have been included and addressed in the discussion.

22. Discussion “However, our competition binding assays show no significant decrease in either affinity or total binding, making the second explanation less likely”. I think your competition assay point into the right direction, even if data (affinity-decrease for agonists in the presence of VUN103) are not significant. This also fits well to the small changes in the BRET assay in Fig5: the presence of agonist causes separation of US28 and the nanobody. In turn, occupancy of the receptor by the nanobody should stabilize a conformation that the agonist radioligand does not really like. I think this is what you see as a trend in your binding assays, but it is difficult to state without statistical significance.

We agree with the reviewer and also considered this a possible explanation. We have performed additional BRET experiments using fluorescently labeled CX3CL1 and CCL5 and different NanoLuc-tagged US28-Nb fusion constructs. We see no decrease in ligand binding affinity for the VUN103-stabilized US28 conformation (Figure 4C-D). However, the BRET data (Figure 5C-D and Figure S8) and IPx data (Figure 6A) show a small but significant effect on VUN103 binding upon addition of CX3CL1. This is consistent with a preferred conformation of US28 seen by VUN103. The differences between the ligand-bound and apo-conformation are indeed small. Moreover, an additional IPx experiment (Figure 6A) showed that co-expression of Nb7 and VUN103 results in full inhibition of US28 signaling, both in the absence and presence of CX3CL1. This is again consistent with recognition of different conformations by the two intrabodies. We have added this data to the manuscript.

23. Discussion, lines 337,338: “The effect of ligand binding on VUN103 binding and inhibition is small but significant.” Maybe better to rephrase to ‘the effect of VUN103 on ligand binding’ OR ‘the effect of VUN on ligand competition for the agonist-occupied receptor’ or the like...’

We have rephrased this to: “Nb7 and VUN103 recognize different conformational states of US28. Despite the fact that binding of CX3CL1 induces significant changes for both nanobodies, these effects are subtle. The apo-conformation and ligand-bound conformation of US28 differ only slightly. This would not be all that surprising, given that US28 is characterized by an exceptionally high constitutive activity. This is also in line with previous studies where ligand binding could only marginally superactivate US28 signaling^{24, 38}. “

24. Discussion, lines 360,361: “Full inhibition of US28 signaling could provide new therapeutic strategies to treat HCMV-positive tumors or latently-infected cells.” This is a strong claim and I totally agree that data are tempting to make such a claim. However before we speculate about therapeutic strategies based on full inhibition of US28 signaling, it would be important to bring the causal proof that in the spheroid or infection paradigm VUN103 effects are indeed mediated by specific inhibition of US28. I made specific suggestions how to achieve this (see above).

We agree with the reviewer. This is in line with the previous comments. By addressing the specificity of inhibition of US28 by VUN103, we have now established that VUN103-mediated inhibition is indeed US28-specific.

Reviewers' Comments:

Reviewer #2:

Remarks to the Author:

The authors have responded adequately to my comments. I support publication and congratulate the authors with the beautiful work.

Jan Steyaert

Reviewer #3:

Remarks to the Author:

I thank the authors for carefully addressing numerous comments and suggestions which led to significant improvement of their manuscript. I do not wish to make additional comments and have no further concerns.